# Removing Multiple Shortcuts through the Lens of Multi-task Learning

## Abstract

We consider the problem of training an unbiased and accurate model using a biased dataset with multiple biases. This problem is challenging since the multiple biases cause multiple undesirable shortcuts during training, and even worse, mitigating one of them may exacerbate another. To address this challenge, we introduce a novel method connecting the problem to multi-task learning (MTL). Our method divides training data into several groups according to their effects on the model bias and defines each task of MTL as solving the target problem for each group. It in turn trains a single model for all the tasks with a weighted sum of task-wise losses as the training objective, while optimizing the weights as well as the model parameters. At the heart of our method lies the weight adjustment algorithm, which is rooted in a theory of multi-objective optimization and guarantees a Pareto-stationary solution. In addition, we also present a new real-image benchmark with multiple biases, dubbed MultiCelebA, for evaluating debiased training methods under realistic and challenging scenarios. Our method achieved the state of the art on three datasets with multiple biases including MultiCelebA, and demonstrated superior performance on conventional single-bias datasets.

## 1 Introduction

Empirical risk minimization (ERM) (Vapnik, 1999) is currently the gold standard in supervised learning of deep neural networks. However, recent studies (Sagawa et al., 2019; Geirhos et al., 2020) revealed that ERM is prone to taking *undesirable shortcuts* stemming from *spurious correlations* between the target labels and irrelevant attributes. For example, Sagawa et al. (2019) observed how a deep neural network trained to classify bird species relies on the background rather than the bird itself. Such a spurious correlation is often hard to mitigate since the data collection procedure itself is biased towards the correlation.

To resolve this issue, researchers have investigated debiased training algorithms, *i.e.*, algorithms training a model while mitigating spurious correlations (Arjovsky et al., 2019; Bahng et al., 2020; Sagawa et al., 2019; Teney et al., 2021; Tartaglione et al., 2021; Lee et al., 2021; Nam et al., 2020; Liu et al., 2021; Kim et al., 2022). They focus on improving performance on bias-conflicting samples (*i.e.*, samples that disagree with the spurious correlations) to achieve a balance of bias-conflicting and bias-guiding samples (*i.e.*, those agreeing with the spurious correlations) in terms of performance. While these algorithms have shown promising results, they have been evaluated in a limited setting where only a single type of spurious correlation exists in a training dataset.

We advocate that debiased training algorithms should be evaluated under more realistic scenarios with multiple biases. In such scenarios, some samples may align with one bias but may conflict with another, which makes mitigating spurious correlations more challenging. If one only considers the intersection of bias-conflicting samples, *i.e.*, *clean* samples that disagree with all the spurious correlations, the resulting group will be extremely small as illustrated in Figure 1 and result in overfitting consequently. Furthermore, mitigation of one bias often promote another as empirically observed by Li et al. (2023).

In this work, we address the aforementioned challenges through multi-task learning (MTL). First, we divide the entire training set into multiple groups where data of the same group have the same impact on training in terms of the model bias, *i.e.*, guiding to or conflicting with each bias type in the same way, as illustrated in Figure 1. Then, we formulate each task of MTL as solving the target

problem for each group. Unlike the conventional MTL setting, this results in tasks that share the same prediction targets but differ in the distribution of the biased attributes.

For training a single model that deals with all the tasks, our optimization method mitigates biases and alleviates between-task conflicts at the same time via aiming for Pareto optimality, *i.e.*, a state where no task can be further improved without sacrificing others. Our method, derived from a multi-objective optimization (MOO) algorithm (Désidéri, 2012), trains a model to reach Pareto-optimal performance for the aforementioned tasks. To this end, it is designed to dynamically adjust task-wise importance weights so that model parameters converge to a Pareto-stationary point. From another point of view, our method can be interpreted as an optimization process to find a flat minimum of the loss landscape (Li & Gong, 2021), which has shown to improve the model's generalization capability (Keskar et al., 2017; Dziugaite & Roy, 2017; Jiang et al., 2020; Li & Gong, 2021; Cha et al., 2021).

We also introduce a new multi-bias benchmark along with the new debiased training algorithm. Our benchmark, dubbed MultiCelebA, is a collection of real facial images from CelebA (Liu et al., 2015), and incorporates multiple bias types that are spuriously correlated with a target class. Compared with existing multi-bias datasets composed of synthetic images (Li et al., 2022; 2023), it allows to evaluate debiased training algorithm on more realistic and challenging scenarios.

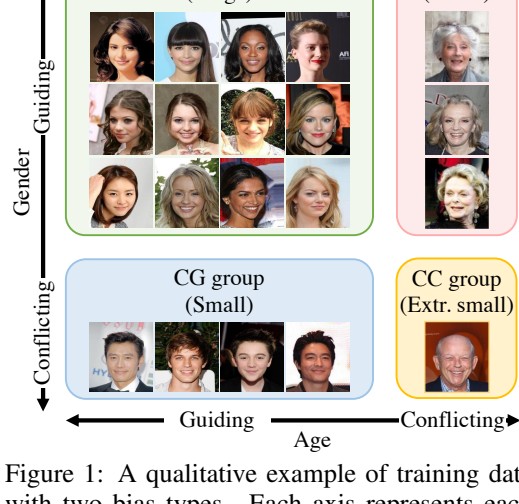

Figure 1: A qualitative example of training data with two bias types. Each axis represents each bias type, for which bias-guiding samples make up the majority and bias-conflicting ones hold the minority. The intersection of bias-conflicting samples (*i.e.*, the smallest group) thus becomes extremely small as the number of bias types increases. To apply MTL, we divide training data into multiple groups so that data of the same group are guiding or conflicting with each bias type in the same way. In this example, the name of each group indicates if samples of the group has a guiding attribute (G) or a conflicting attribute (C) for `gender` and `age`, in respective order.

We extensively evaluated our method on three multi-bias benchmarks including MultiCelebA and three single-bias benchmarks, where it outperformed all the existing debiased training methods. The main contribution of this paper is four-fold:

- This work is the first to interpret debiased training as a MTL problem. Based on this notion, we present a novel and effective debiased training algorithm.
- We present a new real-image multi-bias benchmark for evaluating debiased training methods under realistic and challenging scenarios.
- We benchmarked existing methods for debiased training and demonstrated that they struggle when training data exhibit multiple biases.
- Our method achieved the state of the art on three datasets with multiple biases. In addition, it also showed superior performance on conventional single-bias datasets.

## 2    RELATED WORK

**Debiased training.** A body of research has addressed the bias issue that arises from spurious correlations between target and latent attributes. A group of previous work exploits manual labels for bias attributes (Arjovsky et al., 2019; Bahng et al., 2020; Dhar et al., 2021; Gong et al., 2020; Li & Vasconcelos, 2019; Sagawa et al., 2019; Teney et al., 2021; Tartaglione et al., 2021; Zhu et al., 2021; Zhang et al., 2022; Wang et al., 2018). For instance, Sagawa et al. (2019) presented a robust optimization method that weights groups of different bias attributes differently, Dhar et al. (2021) and Gong et al. (2020) employed adversarial training, and Zhang et al. (2022) proposed using contrastive learning. Later on, debiased training algorithms that do not require any bias supervision

have been studied to reduce the annotation cost (Darlow et al., 2020; Kim et al., 2021; Lee et al., 2021; Nam et al., 2020; Liu et al., 2021; Kim et al., 2022; Hwang et al., 2022). However, whether directly using the bias labels or not, these methods assume that the bias inherent in data is of a single type. This assumption often does not hold in real-world scenarios, where data exhibit multiple biases, and in practice classifiers can be easily biased to multiple independent biases, as shown in StylEx (Lang et al., 2021). Only a few recent studies (Li et al., 2022; 2023) addressed multiple biases with new training algorithms and benchmarks. Li et al. (2022) discovered multiple biases through iterative assignment of pseudo bias labels, while Li et al. (2023) presented an augmentation method that emulates the generation process of bias types. However, their methods are dedicated to handle synthetic images. In contrast, we propose a new algorithm that trains unbiased models regardless of the number and types of biases, along with a new natural image dataset for evaluating debiased training methods in the presence of multiple biases.

**Multi-task learning.** MTL is a research area that aims at developing a single model capable of performing multiple tasks simultaneously. MTL models in general demonstrate superior results to task-specific models by leveraging representation and inductive bias shared across multiple tasks (Ruder, 2017; Sener & Koltun, 2018; Meyerson & Miikkulainen, 2020; Vandenhende et al., 2021). Some of previous work tackle MTL via MOO (Miettinen, 1999; Ehrgott, 2005; Désidéri, 2012), which aims at resolving conflicts between competing objectives in optimization perspectives (Sener & Koltun, 2018; Lin et al., 2019; Wang et al., 2020; Yu et al., 2020; Li & Gong, 2021). Specifically, Li & Gong (2021) proposed a MOO method for training multilingual models to balance the losses between high and low-resource languages. Inspired by these methods, we propose the first algorithm that connects debiasing to MOO. Our work is however different from the conventional MTL approach in terms of task definition, and we present a novel loss tailored to the debiasing problem.

**Fairness with MTL.** Fairness is a research topic related to debiased training, as both share the goal of developing an unbiased model regarding hidden attributes. However, its focus lies in addressing the model bias issues that arise not from spurious correlations but rather from the limited availability of samples in specific domains (*i.e.*, protected attributes). Previous work has addressed fairness concerns through MTL (Oneto et al., 2019) or considered fairness within MTL (Wang et al., 2021). These studies assume a particular model architecture composed of shared and task-specific modules. On the other hand, Maheshwari & Perrot (2022) proposed weighting for fairness, which is broadly considered as MOO. However, these algorithms are not explicitly designed from the view of MOO. In contrast, our method incorporates a single model of an arbitrary architecture, and addresses a single debiasing task from an MTL perspective, with a focus on mitigating spurious correlations.

## 3 PROPOSED METHOD

We propose a novel debiased training algorithm that effectively addresses one or multiple spurious correlations based on a theory of MOO (Désidéri, 2012), assuming that bias attributes are annotated for training data. Our algorithm divides training data into several groups according to their effects on the model bias, defines each task of MTL as solving the target problem for each group, and trains a single model for all the tasks while optimizing importance weights of the tasks as well as the model parameter. The rest of this section first introduces the MOO theory that motivates our work (Section 3.1) and then describes the proposed algorithm in detail (Section 3.2).

### 3.1 PRELIMINARY: MTL AS MOO

We consider MTL as a problem to optimize a parameter $\theta$ with respect to a collection of task-wise training loss functions $L(\theta) = [\mathcal{L}_1(\theta), \ldots, \mathcal{L}_N(\theta)]^\top$. To solve such a problem, MOO frameworks aim at finding a solution that achieves Pareto optimality, *i.e.*, a state where no objective can be improved without sacrificing others.

**Definition 1 (Pareto optimality)** *A parameter $\theta^*$ is Pareto-optimal if there exists no other parameter $\theta$ such that $\mathcal{L}_n(\theta) \leq \mathcal{L}_n(\theta^*)$ for $n = 1, \ldots, N$ and $L(\theta) \neq L(\theta^*)$.*

However, finding the Pareto-optimal parameter is intractable for non-convex loss functions like the training objective of deep neural networks. Instead, one may consider using gradient-based optimization to find a parameter satisfying Pareto stationarity (Désidéri, 2012), *i.e.*, a state where a

convex combination of task-wise gradients equals a zero-vector. Pareto stationarity is a necessary condition for Pareto optimality if the loss functions in $L(\theta)$ are smooth (Désidéri, 2012).

**Definition 2 (Pareto stationarity)** *A parameter $\theta^*$ is Pareto-stationary if there exists a task-scaling vector $\boldsymbol{\alpha} = [\alpha_1, \ldots, \alpha_N]^\top$ satisfying the following condition:*

$$\boldsymbol{\alpha}^\top \nabla_\theta L(\theta^*) = \mathbf{0}, \quad \boldsymbol{\alpha} \geq \mathbf{0}, \quad \boldsymbol{\alpha}^\top \mathbf{1} = 1, \tag{1}$$

*where $\mathbf{0} = [0, \ldots, 0]^\top \in \mathbb{R}^N$ and $\mathbf{1} = [1, \ldots, 1]^\top \in \mathbb{R}^N$.*

Désidéri (2012) proposed the multi-gradient descent algorithm (MGDA) to search for a Pareto-stationary parameter. MGDA finds a task-scaling parameter $\boldsymbol{\alpha}$ which combines the task-wise gradients $\nabla_\theta L$ to be approximately a zero vector by solving the following optimization problem:

$$\min_{\boldsymbol{\alpha}} \left\| \boldsymbol{\alpha}^\top \nabla_\theta L \right\|_2^2, \quad \boldsymbol{\alpha} \geq \mathbf{0}, \quad \boldsymbol{\alpha}^\top \mathbf{1} = 1. \tag{2}$$

Given $\boldsymbol{\alpha}$, MGDA performs a gradient-based update on the parameter $\theta$ with respect to $\boldsymbol{\alpha}^\top L(\theta)$.

## 3.2 DEBIASED TRAINING BY MOO

Our debiased training algorithm based on MOO aims to balance performance on samples with bias-guiding and bias-conflicting attributes. The key idea is to formulate each objective of MOO as optimizing over a group of training samples that have the same impact on training in terms of the model bias. To this end, we partition the entire training set into multiple groups according to the existence of bias-guiding or bias-conflicting attributes. The remainder of this section elaborates on the grouping strategy and the MOO formulation for debiased training.

### 3.2.1 GROUPING STRATEGY

As illustrated in Figure 1, we divide training data into multiple groups so that all data in the same group have the same impact on training in terms of the model bias. To be specific, we consider training a classifier on a dataset $\mathcal{D} = \{(x^{(m)}, t^{(m)})\}_{m=1}^M$, where each sample $x^{(m)}$ is associated with a target class $t^{(m)}$ and a list of attributes $\boldsymbol{b}^{(m)} = [b_1^{(m)}, \ldots, b_D^{(m)}]^\top$. We group the samples using a list of binary group labels $\boldsymbol{g}^{(m)} = [g_1^{(m)}, \ldots, g_D^{(m)}]$ based on whether each attribute $b_d^{(m)}$ is the *majority attribute* in target class $t^{(m)}$, *i.e.*, $g_d^{(m)} = 1$ if

$$b_d^{(m)} = \operatorname*{argmax}_{b_d} \left| \left\{ m' | t^{(m')} = t^{(m)}, b_d^{(m')} = b_d \right\} \right|,$$

and $g_d^{(m)} = 0$ otherwise. This results in $2^D$ groups where samples in the same group share the same group labels. This grouping policy differs from prior work (Sagawa et al., 2019; Kirichenko et al., 2022; Nam et al., 2022; Sagawa et al., 2020; Zhang et al., 2022) that uses the target classes and the attributes as the group labels: each group in our method contains samples from all the target classes, while existing ones only keep a group of samples with the same target class and the same attributes. Hence, our grouping policy enables to conduct the target classification task on each group, and the discrepancy between the groups in spurious correlations prevents a single model trained on all the groups from taking undesirable shortcuts.

We remark that our grouping strategy can be interpreted as a MTL problem, where the tasks share the same target classes but are defined on different groups of samples. Our goal is to train a model capable of accurately classifying samples from all the groups, *i.e.*, its performance should not be biased towards a certain group. Similar to MTL, minimizing a linear combination of group-wise loss functions in a naïve way leads to conflicts between bias-guiding and bias-conflicting groups.

### 3.2.2 TRAINING ALGORITHM

Based on the grouping strategy, we propose an algorithm to optimize over $N = 2^D$ groups while minimizing the conflict between group-wise loss functions. Let $L(\theta) = [\mathcal{L}_1(\theta), \ldots, \mathcal{L}_N(\theta)]^\top$ denote the list of empirical risk functions on $N$ groups and consider minimizing their convex combination $\boldsymbol{\alpha}^\top L(\theta)$ where $\boldsymbol{\alpha} \geq \mathbf{0}$ and $\boldsymbol{\alpha}^\top \mathbf{1} = 1$. To address between-group conflicts, we propose adjusting the group-scaling parameter $\boldsymbol{\alpha}$ such that the training converges to a Pareto-stationary point with a flat loss landscape.

To be specific, our goal is to minimize the training objective $\boldsymbol{\alpha}^\top L(\theta)$ while simultaneously adjusting the group-scaling parameter $\boldsymbol{\alpha}$ to minimize the objective in Eq. (2). To this end, we optimize the following loss function with respect to both $\theta$ and $\boldsymbol{\alpha}$ simultaneously:

$$\hat{L}(\theta) = \boldsymbol{\alpha}^\top L(\theta) + \lambda \left\| \boldsymbol{\alpha}^\top (\nabla L(\theta))_\dagger \right\|_2^2, \quad (3)$$

where $\boldsymbol{\alpha} \geq \mathbf{0}$, $\boldsymbol{\alpha}^\top \mathbf{1} = 1$, $(\cdot)_\dagger$ denotes the stop-gradient operator, and $\lambda$ is a Lagrangian multiplier for the Pareto stationarity objective in Eq. (2). In practice, we re-parameterize group-scaling parameter using a softmax function, *i.e.*,

---

**Algorithm 1:** Debiased training by MOO

**while** *not converged* **do**
    Let $\boldsymbol{\alpha} = \mathrm{SoftMax}(\bar{\boldsymbol{\alpha}})$.
    **for** $u \leftarrow 1$ *to* $U - 1$ **do**
        |   Update $\theta \leftarrow \theta - \eta_1 \boldsymbol{\alpha}^\top \nabla_\theta L(\theta)$.
    **end for**
    Let $\hat{L}(\theta) = \boldsymbol{\alpha}^\top L(\theta) + \lambda \left\| \boldsymbol{\alpha}^\top \nabla_\theta L(\theta) \right\|_2^2$.
    Update $\theta \leftarrow \theta - \eta_1 \boldsymbol{\alpha}^\top \nabla_\theta L(\theta)$.
    Update $\bar{\boldsymbol{\alpha}} \leftarrow \bar{\boldsymbol{\alpha}} - \eta_2 \nabla_{\bar{\boldsymbol{\alpha}}} \hat{L}(\theta)$.
    Update $\lambda \leftarrow \lambda + \eta_2 \nabla_\lambda \hat{L}(\theta)$.
**end while**

---

$\boldsymbol{\alpha} = \mathrm{SoftMax}(\bar{\boldsymbol{\alpha}})$. This allows optimizing over $\bar{\boldsymbol{\alpha}}$ with gradient-based updates without violating the constraints $\boldsymbol{\alpha} \geq \mathbf{0}$ and $\boldsymbol{\alpha}^\top \mathbf{1} = 1$ in Eq. (2). We update the group-scaling parameter $\boldsymbol{\alpha}$ with gradient descent and the Lagrangian multiplier $\lambda$ with gradient ascent every $U$ iterations. The learning process of our method is described in Algorithm 1.

We found that optimizing $\boldsymbol{\alpha}$ to minimize not only the weighted sum of group-wise gradients (*i.e.*, MOO) but also the weighted sum of group-wise losses helps improve overall performance. This is probably because minimizing the weighted sum of group losses enables to balance the group-wise losses of different scales caused by the large discrepancy in size between the groups. Empirical analysis on our training algorithm is provided in the Appendix A.2.

We also note that our method can be interpreted as curvature aware training (Li & Gong, 2021), where the task-scaling parameter $\boldsymbol{\alpha}$ is adjusted to exhibit better generalization for each task. Specifically, Li & Gong (2021) consider adjusting the training objective $\left\| \boldsymbol{\alpha}^\top (\nabla L(\theta)) \right\|_2^2$ so that gradient-based optimization of the parameter $\theta$ converges to a *flat* minimum with a small curvature, *i.e.*, a parameter with a small trace of the Hessian matrix with respect to the training objective. It has been reported in the literature (Keskar et al., 2017; Dziugaite & Roy, 2017; Jiang et al., 2020) that a model converging to such a flat minimum in training has better generalization capability.

### 3.3 DISCUSSION: ON THE USE OF BIAS LABELS

Bias attribute labels would be expensive particularly in the multi-bias setting. However, regarding that debiasing in this setting has been rarely studied so far and is extremely challenging, we believe it is premature to tackle the task in an unsupervised fashion at this time. As in the single bias setting where the society has first developed supervised debiasing methods and then unsupervised counterparts, our algorithm will be a cornerstone of follow-up unsupervised methods in the multi-bias setting. Moreover, the annotation cost for bias labels can be substantially reduced by incorporating existing techniques for pseudo labeling of bias attributes (Jung et al., 2021; Nam et al., 2022).

## 4 MULTICELEBA BENCHMARK

We present a new benchmark, dubbed Multi-CelebA, for evaluating debiased training algorithms under the presence of multiple biases. Unlike Multi-Color MNIST (Li et al., 2022) and UrbanCars (Li et al., 2023) built for the same purpose using synthetic images, MultiCelebA is composed of natural facial images, making it more suitable for simulating real-world scenarios.

MultiCelebA is built upon CelebA (Liu et al., 2015), a large-scale collection of facial images each with 40 attribute annotations. Among these attributes, `high-cheekbones` is chosen as the target class, while `gender`, `age`, and `mouth slightly open` are used as bias attributes that

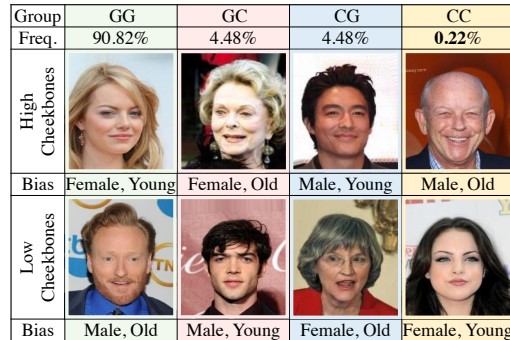

Figure 2: Training set configuration of Multi-CelebA in the two biases setting.

Table 1: Performance in GG, GC, CG, CC, UNBIASED, WORST, and INDIST (%) on MultiCelebA in two biases setting. The first element of each of the four combinations {GG, GC, CG, CC} is about the bias type `gender`, while the second is about the bias type `age`. We mark the best and the second-best performance in **bold** and underline, respectively.

| Method | Bias label | GG | GC | CG | CC | UNBIASED | WORST | INDIST |
|--------|:----------:|----|----|----|----|----------|-------|--------|
| ERM | ✗ | $\mathbf{98.2}_{\pm0.7}$ | $\mathbf{89.2}_{\pm2.6}$ | $58.2_{\pm3.0}$ | $19.0_{\pm1.8}$ | $63.8_{\pm1.2}$ | $14.7_{\pm4.8}$ | $\mathbf{97.0}_{\pm0.2}$ |
| LfF | ✗ | $79.8_{\pm2.6}$ | $71.7_{\pm2.2}$ | $80.2_{\pm1.7}$ | $71.5_{\pm3.3}$ | $75.8_{\pm0.5}$ | $66.8_{\pm1.2}$ | $81.9_{\pm3.1}$ |
| JTT | ✗ | $76.1_{\pm5.2}$ | $60.8_{\pm5.2}$ | $65.1_{\pm10.7}$ | $51.9_{\pm1.6}$ | $64.7_{\pm3.2}$ | $50.1_{\pm2.0}$ | $78.7_{\pm6.5}$ |
| DebiAN | ✗ | $64.4_{\pm30.4}$ | $63.6_{\pm22.2}$ | $49.8_{\pm7.6}$ | $45.5_{\pm13.2}$ | $55.8_{\pm11.7}$ | $25.7_{\pm6.0}$ | $66.8_{\pm34.1}$ |
| Upsampling | ✓ | $79.8_{\pm1.5}$ | $81.0_{\pm1.30}$ | $76.7_{\pm1.1}$ | $75.6_{\pm1.2}$ | $78.3_{\pm0.8}$ | $71.5_{\pm2.0}$ | $82.6_{\pm0.8}$ |
| Upweighting | ✓ | $79.0_{\pm4.1}$ | $79.2_{\pm6.02}$ | $\underline{80.8}_{\pm0.0}$ | $\underline{78.7}_{\pm3.6}$ | $\underline{79.4}_{\pm3.4}$ | $\underline{73.5}_{\pm4.2}$ | $83.4_{\pm5.9}$ |
| GroupDRO | ✓ | $81.2_{\pm1.0}$ | $81.2_{\pm1.2}$ | $76.7_{\pm1.5}$ | $74.6_{\pm0.4}$ | $78.4_{\pm0.7}$ | $71.6_{\pm1.1}$ | $83.5_{\pm0.7}$ |
| SUBG | ✓ | $77.1_{\pm1.0}$ | $78.4_{\pm0.7}$ | $77.5_{\pm1.7}$ | $78.0_{\pm1.2}$ | $77.7_{\pm0.6}$ | $69.6_{\pm0.7}$ | $80.3_{\pm1.1}$ |
| LISA | ✓ | $82.8_{\pm1.3}$ | $83.2_{\pm0.5}$ | $79.8_{\pm0.8}$ | $77.6_{\pm2.6}$ | $\underline{80.9}_{\pm0.2}$ | $72.8_{\pm1.5}$ | $84.5_{\pm1.7}$ |
| DFR$_{tr}^{tr}$ | ✓ | $\underline{91.3}_{\pm3.5}$ | $83.6_{\pm4.0}$ | $46.7_{\pm3.8}$ | $28.5_{\pm4.6}$ | $62.5_{\pm0.6}$ | $12.3_{\pm8.5}$ | $\underline{85.5}_{\pm6.2}$ |
| Ours | ✓ | $82.4_{\pm0.9}$ | $\underline{85.1}_{\pm0.4}$ | $\mathbf{81.7}_{\pm0.4}$ | $\mathbf{82.6}_{\pm1.0}$ | $\mathbf{82.9}_{\pm0.2}$ | $\mathbf{77.9}_{\pm0.2}$ | $84.3_{\pm0.9}$ |

are spuriously correlated with `high-cheekbones` and thus cause undesirable shortcuts during training. Note that these bias attributes are not randomly chosen but identified by following the empirical analysis of Scimeca et al. (2022), which revealed that these attributes are strongly correlated with the target class; details of the analysis are presented in Appendix A.3.

Based on MultiCelebA, we present two different benchmark settings: one with two bias attributes `gender` and `age`, and the other with all the three bias attributes. In both of the two settings, to simulate challenging scenarios where training data are extremely biased, we set the bias-guiding samples for each bias type to $95.3\%$ so that only $0.22\%$ of training samples are free from spurious correlations in the two biases setting and $0.07\%$ for the three biases settings. Example images and the frequency of each attribute in the two biases setting are presented in Figure 2.

# 5 EXPERIMENTS

## 5.1 SETUP

**Datasets.** We adopt three multi-bias benchmarks, MultiCelebA, UrbanCars (Li et al., 2023), and Multi-Color MNIST (Li et al., 2022), and three single-bias datasets, Waterbirds (Sagawa et al., 2019), CelebA (Liu et al., 2015), and BFFHQ (Lee et al., 2021), for evaluation.

**Evaluation metrics.** The quality of debiased training algorithms is measured mainly by UNBIASED, the average of group average accuracy scores. For the benchmarks with two bias types, we also adopt average accuracy for each of the four groups categorized by the guiding or conflicting nature of the biases: {GG, GC, CG, CC}, where G and C indicate whether a group includes bias-guiding or bias-conflicting samples for each bias type, respectively. We also report WORST, the minimum group average accuracy, and INDIST, the weighted average of group accuracy scores where the weights are proportional to group sizes (Sagawa et al., 2019).

**Baselines.** We compare our algorithm with a large body of existing debiased training algorithms. Among them, GroupDRO (Sagawa et al., 2019), SUBG (Sagawa et al., 2020), LISA (Yao et al., 2022), and DFR Kirichenko et al. (2022) as well as simple upsampling and upweighting strategies demand true bias labels of training data like ours, while LfF (Nam et al., 2020), JTT (Liu et al., 2021), EIIL (Creager et al., 2021), PGI (Ahmed et al., 2021), and DebiAN (Li et al., 2022) do not.

**Implementation details.** Following previous work, we conduct experiments using different neural network architectures for different datasets: a three-layered MLP for Multi-Color MNIST, ResNet18 for MultiCelebA and BFFHQ, and ResNet50 for UrbanCars, Waterbirds, and CelebA. The group-scaling parameter $\alpha$ is initialized to $\frac{1}{N}\mathbf{1}$ where $N$ is the number of groups, and the Lagrangian multiplier $\lambda$ is initialized to 0. For mini-batch construction during training, group-balanced sampling is used to compute each loss for multiple tasks. We report the average and standard deviation of each metric calculated from three runs with different random seeds. More implementation details are provided in Appendix A.4.

Table 2: Performance in UNBIASED, WORST, and INDIST (%) on MultiCelebA in three biases setting. We mark the best and the second-best performance in **bold** and underline, respectively.

| Method | UNBIASED | WORST | INDIST |
|---|---|---|---|
| ERM | $64.1_{\pm 0.6}$ | $12.0_{\pm 1.2}$ | $\mathbf{96.7}_{\pm 0.2}$ |
| LfF | $71.7_{\pm 0.6}$ | $47.7_{\pm 5.5}$ | $81.2_{\pm 2.9}$ |
| Upsampling | $72.2_{\pm 0.7}$ | $60.2_{\pm 3.2}$ | $81.6_{\pm 0.1}$ |
| Upweighting | $\underline{78.2}_{\pm 1.3}$ | $\mathbf{65.8}_{\pm 3.0}$ | $82.7_{\pm 1.6}$ |
| GroupDRO | $74.8_{\pm 1.8}$ | $60.4_{\pm 2.8}$ | $\underline{86.1}_{\pm 2.0}$ |
| LISA | $75.6_{\pm 0.4}$ | $61.6_{\pm 2.2}$ | $85.0_{\pm 0.4}$ |
| Ours | $\mathbf{78.9}_{\pm 0.3}$ | $\underline{65.2}_{\pm 2.2}$ | $84.2_{\pm 0.8}$ |

Table 3: Performance in INDIST and CC (%) on UrbanCars. We mark the best and the second-best in **bold** and underline, respectively.

| Method | Bias label | INDIST | CC | GAP |
|---|---|---|---|---|
| ERM | ✗ | $\mathbf{97.6}$ | 28.4 | -69.2 |
| Upsampling | ✓ | 92.8 | 76.0 | -16.8 |
| Upweighting | ✓ | 93.4 | 80.0 | -13.4 |
| GroupDRO | ✓ | 91.6 | 75.2 | -16.4 |
| SUBG | ✓ | 71.1 | 64.8 | $\underline{-6.3}$ |
| LISA | ✓ | $\underline{94.6}$ | $\underline{80.8}$ | -13.8 |
| DFR$_{tr}^{tr}$ | ✓ | 89.7 | 44.5 | -45.2 |
| Ours | ✓ | 91.8 | $\mathbf{87.6}$ | $\mathbf{-4.2}$ |

Table 4: Performance in GG, GC, CG, CC, and UNBIASED (%) on Multi-Color MNIST. The first element of each of the four combinations {GG, GC, CG, CC} is about the bias type `left-color`, while the second is about the bias type `right-color`. We mark the best and the second-best performance in **bold** and underline, respectively.

| Method | Bias label | GG | GC | CG | CC | UNBIASED |
|---|---|---|---|---|---|---|
| ERM | ✗ | $100.0_{\pm 0.0}$ | $\underline{96.5}_{\pm 1.2}$ | $79.5_{\pm 2.5}$ | $20.8_{\pm 1.1}$ | $74.2_{\pm 1.1}$ |
| LfF | ✗ | $99.6_{\pm 0.5}$ | $4.7_{\pm 0.5}$ | $\mathbf{98.6}_{\pm 0.4}$ | $5.1_{\pm 0.4}$ | $52.0_{\pm 0.1}$ |
| EIIL | ✗ | $100.0_{\pm 0.0}$ | $\mathbf{97.2}_{\pm 1.5}$ | $70.8_{\pm 4.9}$ | $10.9_{\pm 0.8}$ | $69.7_{\pm 1.0}$ |
| PGI | ✗ | $98.6_{\pm 2.3}$ | $82.6_{\pm 19.6}$ | $26.6_{\pm 5.5}$ | $9.5_{\pm 3.2}$ | $54.3_{\pm 4.0}$ |
| DebiAN | ✗ | $100.0_{\pm 0.0}$ | $95.6_{\pm 0.8}$ | $76.5_{\pm 0.7}$ | $16.0_{\pm 1.8}$ | $72.0_{\pm 0.8}$ |
| Upsampling | ✓ | $99.4_{\pm 0.6}$ | $89.8_{\pm 1.4}$ | $81.3_{\pm 2.6}$ | $42.0_{\pm 1.7}$ | $78.1_{\pm 1.4}$ |
| Upweighting | ✓ | $100.0_{\pm 0.0}$ | $90.0_{\pm 2.5}$ | $\underline{83.4}_{\pm 2.1}$ | $37.1_{\pm 2.8}$ | $77.6_{\pm 1.0}$ |
| GroupDRO | ✓ | $98.0_{\pm 0.0}$ | $87.2_{\pm 4.3}$ | $77.3_{\pm 7.5}$ | $\mathbf{52.3}_{\pm 2.6}$ | $\underline{78.7}_{\pm 2.7}$ |
| Ours | ✓ | $99.7_{\pm 0.6}$ | $90.4_{\pm 3.4}$ | $81.8_{\pm 4.0}$ | $\underline{48.1}_{\pm 0.3}$ | $\mathbf{80.0}_{\pm 2.0}$ |

## 5.2 QUANTITATIVE RESULTS

**MultiCelebA in two biases setting.** In Table 1, we present the results of our experiments evaluating the performance of various baselines and existing debiased training methods on MultiCelebA. One can observe how our method outperforms the baselines by a significant margin in UNBIASED, CG, CC, and WORST metrics. Our method even achieves a second-best accuracy in the GC metric and a moderate accuracy in the GG metric. This highlights how our method successfully prevents performance degradation by simultaneously removing multiple spurious correlations. Intriguingly, we observe that algorithms like JTT, DebiAN, and DFR exhibit UNBIASED metric similar or even lower than the vanilla ERM algorithm. Our hypothesis is that this performance degradation stems from conflicts between removal of different spurious correlations. To be specific, JTT (Liu et al., 2021) exhibits varying accuracy across the GG, GC, CG, and CC groups, indicating that the model is biased towards both `gender` and `age` biases. DebiAN (Li et al., 2022) shows high accuracy in the GG and GC groups, but low accuracy in the CG and CC groups, indicating that the algorithm partially mitigates `age` bias but still suffers from `gender` bias. We also observe that DFR (Kirichenko et al., 2022) achieves lower CC and CG metrics than ERM, suggesting that an ERM-based feature representation alone is insufficient in multi-bias setting. The remaining algorithms, *e.g.*, Upsampling, GroupDRO (Sagawa et al., 2019), and LISA (Yao et al., 2022) show overall decent performance, but the GG and GC metrics are slightly higher than that in CG and CC groups, indicating that the model is still biased towards the `gender` attribute. Surprisingly, the upweighting baseline achieved the second-best performance in CG, CC, and WORST on MultiCelebA, surpassing all the existing debiased training methods.

**MultiCelebA in three biases setting.** Results of the experiment with three bias types are reported in Table 2, where our method substantially outperformed existing methods and baselines in UNBIASED. This demonstrates the scalability of our method to more than two bias types. ERM exhibits significantly lower worst accuracy in the three biases setting compared to the two biases setting. This arises as the number of bias types increases, resulting in a substantially reduced size of smallest group, demonstrating a more challenging setting. Upweighting achieved the highest WORST accuracy, but it exhibited a substantial decline in INDIST performance. In constrast, we achieved best UNBIASED accuracy with compatible WROST and high InDist performance, indicating performs well and unbiased for all groups.

Table 5: Comparisons among different strategies for adjusting the group-scaling parameter $\boldsymbol{\alpha}$ on MultiCelebA in two biases setting. (a) Fixing $\boldsymbol{\alpha}$ by $\frac{1}{N}\mathbf{1}$. (b) Minimizing $\boldsymbol{\alpha}^\top L(\theta)$. (c) MGDA. (d) GradNorm. (e) MoCo, the latest MOO method. (f) Ours minimizing $\hat{L}(\theta)$.

| | GG | GC | CG | CC | UNBIASED | WORST | INDIST |
|---|---|---|---|---|---|---|---|
| (a) No optimization | $79.6_{\pm 2.9}$ | $80.0_{\pm 2.2}$ | $79.0_{\pm 1.9}$ | $78.4_{\pm 1.3}$ | $79.2_{\pm 1.4}$ | $70.8_{\pm 2.7}$ | $78.5_{\pm 5.7}$ |
| (b) Minimizing group losses | $76.4_{\pm 2.2}$ | $77.8_{\pm 0.4}$ | $77.1_{\pm 2.2}$ | $78.0_{\pm 1.7}$ | $77.3_{\pm 0.6}$ | $67.6_{\pm 0.3}$ | $81.3_{\pm 3.6}$ |
| (c) MGDA | $81.6_{\pm 3.5}$ | $85.1_{\pm 2.1}$ | $80.1_{\pm 1.3}$ | $82.3_{\pm 3.2}$ | $82.3_{\pm 0.4}$ | $73.9_{\pm 0.2}$ | $82.7_{\pm 3.7}$ |
| (d) GradNorm | $\mathbf{85.9}_{\pm 5.8}$ | $\mathbf{86.9}_{\pm 2.4}$ | $78.1_{\pm 3.5}$ | $76.6_{\pm 6.5}$ | $81.9_{\pm 0.6}$ | $70.9_{\pm 4.9}$ | $\mathbf{86.5}_{\pm 5.4}$ |
| (e) MoCo | $81.7_{\pm 1.3}$ | $81.8_{\pm 2.6}$ | $77.2_{\pm 0.9}$ | $74.9_{\pm 1.2}$ | $78.9_{\pm 1.5}$ | $72.1_{\pm 2.7}$ | $83.8_{\pm 1.6}$ |
| (f) Ours | $82.4_{\pm 0.9}$ | $85.1_{\pm 0.4}$ | $\mathbf{81.7}_{\pm 0.4}$ | $\mathbf{82.6}_{\pm 1.0}$ | $\mathbf{82.9}_{\pm 0.2}$ | $\mathbf{77.9}_{\pm 0.2}$ | $84.3_{\pm 0.9}$ |

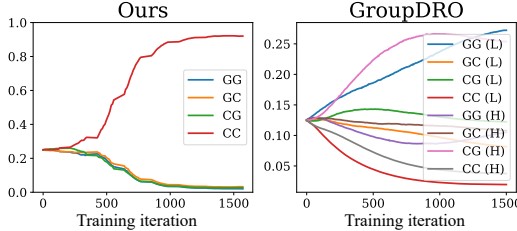

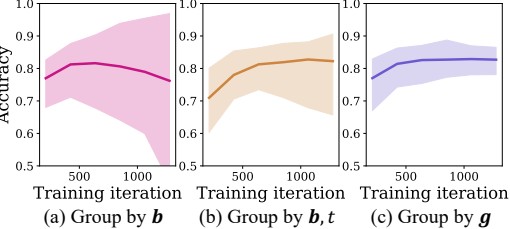

Figure 3: Change of the group-scaling parameter $\boldsymbol{\alpha}$ over time on MultiCelebA in two biases setting. In the case of GroupDRO, (H) and (L) denote `High-cheekbones` and `Low-cheekbones`, respectively.

Figure 4: Group-wise test accuracy of three different grouping strategies. Lines indicate UNBIASED performance, and shaded regions show the lowest (*i.e.*, WORST) and the highest accuracy among the group-wise scores.

**UrbanCars.** In Table 3, we present the results of debiased training algorithms that exploit bias labels and share the identical network structure. Our method achieved significantly superior CC accuracy when compared to method using bias labels, demonstrating a substantial difference.

**Multi-Color MNIST.** In Table 4, we report the evaluation results for the Multi-Color MNIST dataset. Note that we re-use the performance of LfF (Nam et al., 2020), EIIL (Creager et al., 2021), PGI (Ahmed et al., 2021), and DebiAN (Li et al., 2022) reported by Li et al. (2022). Overall, our method demonstrates the best performance along with GroupDRO. In particular, our algorithm exhibits the highest UNBIASED accuracy and the second-best CC accuracy.

**Single-bias datasets.** Surprisingly, our method achieves the best WORST accuracy on Waterbirds and CelebA, and the best UNBIASED on BFFHQ, indicating that our method is effective not only for multi-bias settings but also for single-bias settings. We provide the results in Appendix A.6.

### 5.3 IN-DEPTH ANALYSIS

**Comparisons among different strategies for adjusting $\alpha$.** We first verify the impact of our strategy for adjusting the group-scaling parameter $\alpha$. In Table 5, we compare our training strategy with five alternatives: (a) Using a fixed uniform group-scaling parameter $\boldsymbol{\alpha} = \frac{1}{N}\mathbf{1}$ (*i.e.*, no optimization), (b) minimizing group losses $\boldsymbol{\alpha}^\top L(\theta)$, (c) MGDA that minimizes $\left\| \boldsymbol{\alpha}^\top (\nabla L(\theta))_\dagger \right\|_2^2$, (d) GradNorm (Chen et al., 2018), (e) MoCo, the latest technique for MOO method (Fernando et al., 2023), and (f) our method that minimizes $\hat{L}$ in Eq. (3). Intriguingly, (b) leads to worse performance compared to (a) that uses a fixed value for $\boldsymbol{\alpha}$. We found that utilizing a learnable group-scaling parameter based solely on the weighted sum of group-wise losses resulted in worse performance in all metrics except INDIST when compared with training without it. The results in (c), (d), and (e) demonstrate that blindly applying an existing MOO method as-is with our grouping strategy falls short of the desired level of unbiased performance during training on a biased dataset. This highlights the superiority of our method in scenarios involving multiple spurious correlations.

**Change of the group-scaling parameter $\alpha$ over time.** We compare the trend of group-scaling parameter $\alpha$ in our method with that of GroupDRO (Sagawa et al., 2019) on MultiCelebA in two biases setting, as illustrated in Figure 3. Our method shows an increasing trend for the weight of the CC group, while those of the other groups decrease during training. This indicates that the model

Table 6: Ablation study on the grouping strategy on MultiCelebA in two biases setting: Grouping by bias attribute $b$, grouping by both bias attribute and target class $(b, t)$, and our strategy using the list of binary group labels $g$. SUBG and GroupDRO with our grouping strategy are indicated by †.

| Method | group by | GG | GC | CG | CC | UNBIASED | WORST | INDIST |
|---|---|---|---|---|---|---|---|---|
| ERM | - | $\mathbf{98.2}_{\pm 0.7}$ | $\mathbf{89.2}_{\pm 2.6}$ | $58.2_{\pm 3.0}$ | $19.0_{\pm 1.8}$ | $63.8_{\pm 1.2}$ | $14.7_{\pm 4.8}$ | $\mathbf{97.0}_{\pm 0.2}$ |
| SUBG | $b, t$ | $77.1_{\pm 1.0}$ | $78.4_{\pm 0.7}$ | $77.5_{\pm 1.7}$ | $78.0_{\pm 1.2}$ | $77.7_{\pm 0.6}$ | $69.6_{\pm 0.7}$ | $80.3_{\pm 1.1}$ |
| SUBG† | $g$ | $78.5_{\pm 4.3}$ | $75.9_{\pm 3.3}$ | $71.9_{\pm 2.2}$ | $67.0_{\pm 2.3}$ | $73.3_{\pm 1.7}$ | $63.1_{\pm 6.0}$ | $80.4_{\pm 2.7}$ |
| GroupDRO | $b, t$ | $81.2_{\pm 1.0}$ | $81.2_{\pm 1.2}$ | $76.7_{\pm 1.5}$ | $74.6_{\pm 0.4}$ | $78.4_{\pm 0.7}$ | $\underline{71.6}_{\pm 1.1}$ | $83.5_{\pm 0.7}$ |
| GroupDRO† | $g$ | $\underline{83.1}_{\pm 1.9}$ | $79.5_{\pm 2.4}$ | $\underline{80.7}_{\pm 1.2}$ | $71.8_{\pm 1.1}$ | $78.8_{\pm 1.4}$ | $70.3_{\pm 2.0}$ | $\underline{85.8}_{\pm 1.5}$ |
| Ours | $b$ | $79.5_{\pm 4.6}$ | $79.8_{\pm 3.5}$ | $78.1_{\pm 2.1}$ | $77.0_{\pm 1.6}$ | $78.6_{\pm 2.0}$ | $69.8_{\pm 3.2}$ | $79.2_{\pm 0.7}$ |
| Ours | $b, t$ | $79.4_{\pm 2.9}$ | $80.0_{\pm 2.2}$ | $79.0_{\pm 1.9}$ | $\underline{78.5}_{\pm 1.3}$ | $\underline{79.2}_{\pm 1.4}$ | $71.0_{\pm 2.7}$ | $78.5_{\pm 5.5}$ |
| Ours | $g$ | $82.4_{\pm 0.9}$ | $\underline{85.1}_{\pm 0.4}$ | $\mathbf{81.7}_{\pm 0.4}$ | $\mathbf{82.6}_{\pm 1.0}$ | $\mathbf{82.9}_{\pm 0.2}$ | $\mathbf{77.9}_{\pm 0.2}$ | $84.3_{\pm 0.9}$ |

initially learns a shared representation that incorporates information from all the groups, but later focuses more on the minority group. On the other hand, GroupDRO exhibits a decreasing weight trend for the minority groups (CC (L) and CC (H) in Figure 3). This trend occurs because the minority groups have lower training losses in the early stages of training, leading to lower weights in GroupDRO. As a consequence, it tends to ignore minority groups and exacerbate the bias issue, resulting in inferior performance compared to the upweigthing baseline as shown in Table 1.

**Ablation study on the grouping strategy.** To verify the contribution of our grouping strategy, we compare ours with two alternatives: grouping samples assigned the same bias attribute $b$, and grouping those with the same pair of bias attribute $b$ and target class $t$. Figure 4 demonstrates performance variations by different grouping policies. Figure 4(a) shows that the test accuracy gap between groups enlarges as training progresses when using the bias attribute grouping. We conjecture that this problem arises from class imbalance within the groups categorized solely by bias attributes. Specifically, the number of samples belonging to a target class that is spuriously correlated with the bias attribute becomes dominant, leading to an imbalanced representation of target classes within the group. In Figure 4(b), we applied the commonly used strategy: grouping by both target classes and bias attributes. Compared with the conventional grouping, our method demonstrates a smaller performance gap between groups and higher worst group accuracy, as shown in Figure 4(c). Finally, we also report the performance metrics in Table 6, which demonstrates that our grouping strategy outperforms the others in four metrics.

**Applying our grouping strategy to GroupDRO and SUBG.** We compare our method with Group-DRO and SUBG using the same grouping strategy in Table 6. Results in the table suggest that applying our grouping strategy alone to existing debiased training methods failed to achieve performance comparable to ours. This highlights the contribution of both our debiased training algorithm and grouping strategy to performance improvement.

**Impact of the update frequency $U$.** We conducted experiments to examine how hyperparameter $U$ affects the performance of our method. Table 7 reports the performance in GG, GC, CG, CC and UN-BIASED metrics on MultiCelebA using five different values of $U$. To disregard the influence of the learning rate $\eta_2$, we adjusted the learning rate $\eta_2$ inversely proportional to the increase in the value of $U$. We found that the UNBIASED remained consistent across all $U$ values we examine, which suggests that our algorithm is not sensitive to $U$.

Table 7: Impact of the update frequency $U$ of the group-scaling parameter $\alpha$ on MultiCelebA in two biases setting.

| $U$ | GG | GC | CG | CC | UNBIASED |
|---|---|---|---|---|---|
| 1 | $84.2_{\pm 0.5}$ | $86.0_{\pm 0.5}$ | $80.8_{\pm 0.5}$ | $80.8_{\pm 0.5}$ | $82.9_{\pm 0.3}$ |
| 5 | $83.3_{\pm 0.4}$ | $85.8_{\pm 0.7}$ | $81.2_{\pm 0.4}$ | $81.7_{\pm 0.1}$ | $83.0_{\pm 0.1}$ |
| 10 | $82.9_{\pm 0.2}$ | $82.4_{\pm 0.6}$ | $85.1_{\pm 0.4}$ | $81.7_{\pm 0.3}$ | $82.6_{\pm 0.9}$ |
| 20 | $82.9_{\pm 0.3}$ | $81.9_{\pm 0.5}$ | $84.9_{\pm 0.5}$ | $81.8_{\pm 0.5}$ | $83.0_{\pm 0.9}$ |
| 30 | $79.3_{\pm 1.3}$ | $84.0_{\pm 0.2}$ | $82.6_{\pm 0.3}$ | $85.0_{\pm 0.8}$ | $82.7_{\pm 0.2}$ |

## 6 CONCLUSION

We have presented a novel debiased training algorithm that addresses the challenges posed by multiple biases in training data, inspired by multi-task learning (MTL). In addition, we have introduced a new real-image multi-bias dataset, dubbed MultCelebA. Our method surpassed existing algorithms for debaised training in both multi-bias and single-bias settings on six benchmarks in total. We believe that our work opens a new research direction that connects debiasing and MTL, and will facilitate future research on debiasing under more realistic and challenging scenarios.

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

# A  APPENDIX

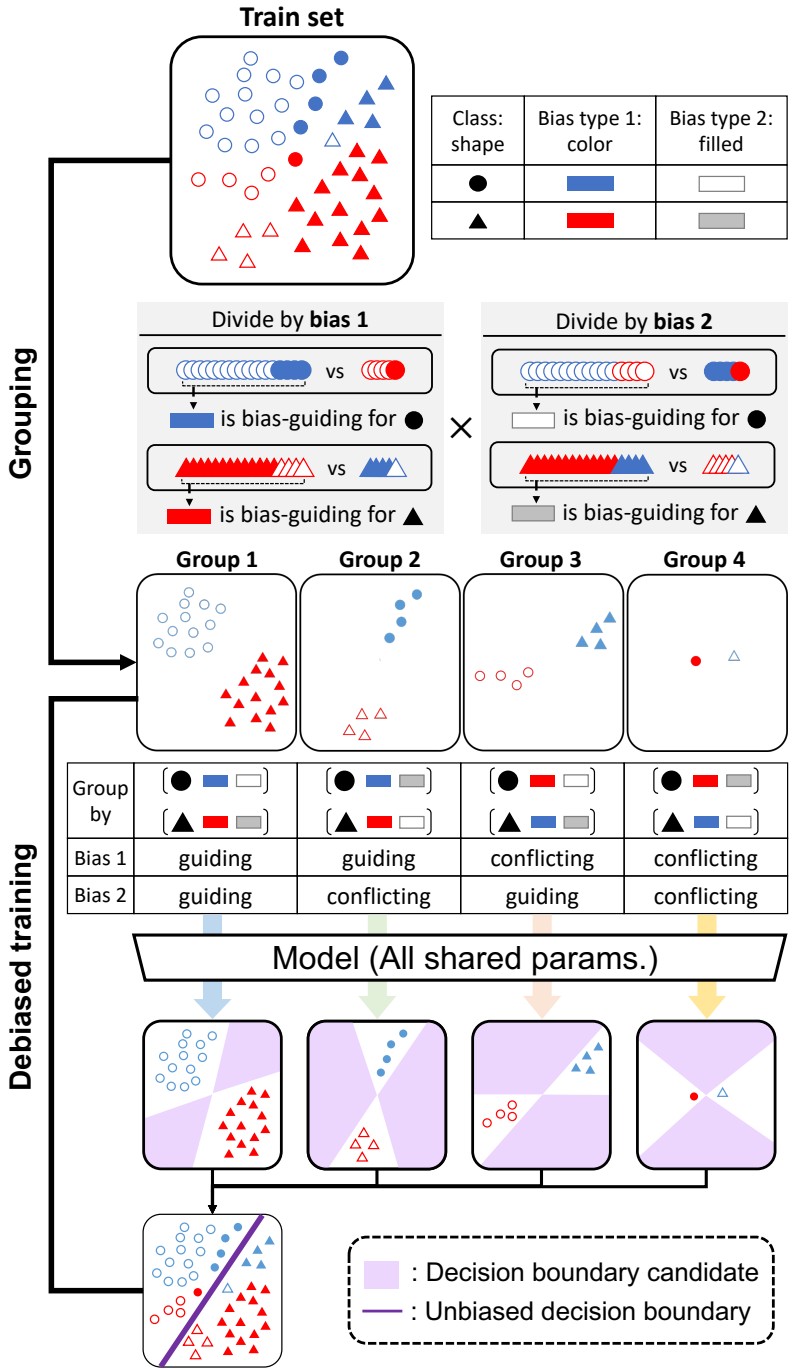

Figure A1: Overview of our method. 1. **Our grouping strategy**: For each sample, its shape means its class while its color and pattern indicate its attributes for two different bias types, respectively. For each class and each bias type, we examine which bias attribute is spuriously correlated with the class and induces the model bias in consequence. Samples that guide to or conflict with each bias type in the same way are grouped together. 2. **Debiased training through the lens of MTL**: The target classification on each group is considered as each task of MTL. We aim to train a model towards a consensus on decision boundary for all the tasks and rely solely on the target-relevant features.

### A.1 OVERVIEW OF OUR METHOD

We introduce our method step by step, which is illustrated in Figure A1. Our method consists of two parts, grouping and debiased training.

**Grouping.** We divide training data into multiple groups so that all data in the same group have the same impact on training in terms of the model bias. First, we define bias-guiding and bias-conflicting attribute of each target class in each bias types. Next, we define groups for each combination of bias-guiding and bias-conflicting labels. For example, bias-guiding for bias type 1 and bias-guiding for bias type 2 is group 1 (GG group), bias-guiding for bias type 1 and bias-conflicting for bias type 2 is group 2 (GC group), bias-conflicting for bias type 1 and bias-guiding for bias type 2 is group 3 (CG group), and bias-conflicting for bias type 1 and bias-conflicting for bias type 2 is group 3 (CC group). Finally, we divide train data into these defined groups. Our grouping strategy enables the target classification task on each group, and the discrepancy between the groups in spurious correlations prevents a single model trained on all the groups from taking undesirable shortcuts.

**Debiased training.** Train set grouping enables us to conduct debiased training using a MTL approach. Unlike conventional MTL, our network does not require task-specific parameters; all parameters are shared. Each sample is forwarded to the model, and we calculate our objective based on its group label. Throughout training, the model strikes balance between bias-guiding and bias-conflicting groups, ultimately achieving an unbiased decision boundary. For instance, in Figure A1, the purple region illustrates decision boundary candidates for each group, with each decision boundary drawn independently of the other groups. With the exception of the group containing all bias-conflicting labels, these decision boundary candidates include biased decisions that work for specific groups but not for others. Our method's objective is to train a model with a mutually agreed-upon decision boundary among all tasks, ensuring it remains unbiased in the presence of all biases. This unbaised decision boundary is depicted as the solid purple line in Figure A1.

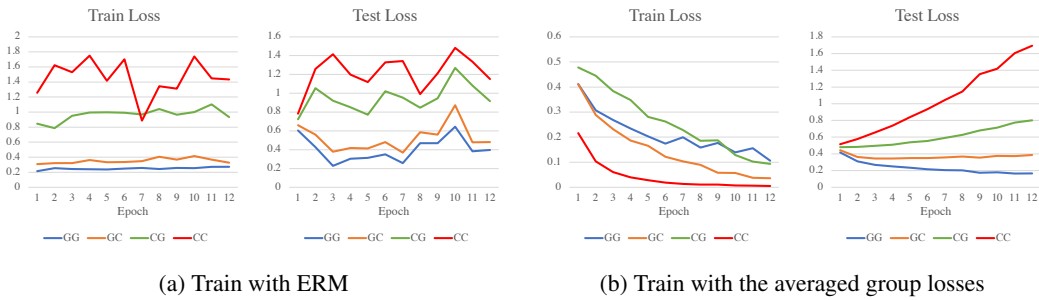

(a) Train with ERM         (b) Train with the averaged group losses

Figure A2: Group losses of (a) model with ERM (b) model with the averaged group losses.

### A.2 EMPIRICAL ANALYSIS OF THE OBJECTIVE FOR OPTIMIZING $\alpha$

When training a model by ERM, the train loss for the small group is larger than that for the large group, and a similar trend is observed in the test loss, as shown in Figure A2(a). Thus, increasing the weight of groups with a larger train loss can be beneficial in giving more weight to the minority group.

However, when we compute the objective by averaging group-wise losses, the gap between train loss and test loss of each group varies depending on group size, as shown in Figure A2(b). This phenomenon arises because smaller groups are more susceptible to memorization effects.

To mitigate the gap between train loss and test loss resulting from memorization effect, Sagawa et al. (2019) has proposed the use of strong regularization on model parameters and an increase in the weight of group with large train loss. This approach of increasing the weight of groups or samples with large train loss has evolved into a standard practice within debiased training methods.

However, in scenarios involving multiple biases, the size of minority group is significantly smaller compared to single bias settings. As a result, much stronger regularization may be required to combat memorization problems. Nevertheless, such strong regularization imposes constraints on

computational capacity and may not fully prevent memorization problems. Increasing the weight according to train loss can ultimately result in a decline in overall performance and the potential exacerbation of model biases. That is the underlying cause of inferior performance of GroupDRO in multiple biases settings compared to Upweighting.

In contrast, our method adjusts weights based on group-wise gradients. Additionally, during the early training stage, the proposed method increases the weight of the group with a small train loss, effectively giving more weight to the minority group.

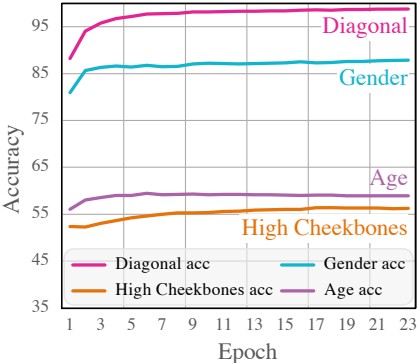

Figure A3: Unbiased accuracy for predicting each attributes

### A.3  BIAS ATTRIBUTES FOR MULTICELEBA

Scimeca et al. (2022) examined how deep neural networks exhibit a preference for attributes based on their easy of learning. Following Scimeca et al. (2022), we assessed the preference of ResNet18 for the target class (`high-cheekbones`) and biases (`gender` and `age`) by evaluating a model trained on diagonal set (GG group in the main paper), where all samples are spuriously correlated with all biases, as shown in Figure A3. Each line on Figure A3 represents unbiased accuracy of a testing attribute, which we used to evaluate the model's ability to predict each testing attribute. ResNet18 exhibited higher unbiased accuracy for `gender` and `age` compared to that of `high-cheekbones`, indicating that the model tends to exploit `gender` and `age` as shortcuts when learning `high-cheekbones` classification task on MultiCelebA. After curating CelebA to MultiCelebA in two biases setting, we additionally introduce `Mouth slightly open` attributes as third bias types. The configurations of MultiCelebA in two biases setting and MultiCelebA in three biases setting are shown in Table A1&A2.

| Group | {Target class, Bias type 1, Bias types 2} | # of train samples |
|---|---|---|
| GG | {High Cheekbones, Female, Young} | 44582 |
| | {Low Cheekbones, Male, Old} | 16220 |
| GC | {High Cheekbones, Female, Old} | 2200 |
| | {Low Cheekbones, Male, Young} | 800 |
| CG | {High Cheekbones, Male, Young} | 2200 |
| | {Low Cheekbones, Female, Old} | 800 |
| CC | {High Cheekbones, Male, Old} | 110 |
| | {Low Cheekbones, Female, Young} | 40 |

Table A1: Configuration of MultiCelebA in two biases setting

### A.4  IMPLEMENTATION DETAILS

#### A.4.1  DATASETS

To evaluate our framework, we consider three multi-bias datasets, *i.e.*, MultiCelebA, Multi-Color MNIST, and UrbanCars and three single-bias datasets, *i.e.*, Waterbirds, CelebA, and BFFHQ. In what follows, we provide details of each dataset.

| Group | {Target class, Bias type 1, Bias types 2, Bias types 3} | # of train samples |
|-------|---------------------------------------------------------|--------------------|
| GGG | {`High Cheekbones`, `Female`, `Young`, `Mouth open`}
{`Low Cheekbones`, `Male`, `Old`, `Mouth close`} | 31491
11336 |
| GGC | {`High Cheekbones`, `Female`, `Young`, `Mouth close`}
{`Low Cheekbones`, `Male`, `Old`, `Mouth open`} | 13091
4884 |
| GCG | {`High Cheekbones`, `Female`, `Old`, `Mouth open`}
{`Low Cheekbones`, `Male`, `Young`, `Mouth close`} | 1594
567 |
| CGG | {`High Cheekbones`, `Male`, `Young`, `Mouth open`}
{`Low Cheekbones`, `Female`, `Old`, `Mouth close`} | 1645
485 |
| GCC | {`High Cheekbones`, `Female`, `Old`, `Mouth close`}
{`Low Cheekbones`, `Male`, `Young`, `Mouth open`} | 606
233 |
| CGC | {`High Cheekbones`, `Male`, `Young`, `Mouth close`}
{`Low Cheekbones`, `Female`, `Old`, `Mouth open`} | 555
315 |
| CCG | {`High Cheekbones`, `Male`, `Old`, `Mouth open`}
{`Low Cheekbones`,`Female`, `Young`, `Mouth close`} | 77
26 |
| CCC | {`High Cheekbones`, `Male`, `Old`, `Mouth close`}
{`Low Cheekbones`,`Female`, `Young`, `Mouth open`} | 33
14 |

Table A2: Configuration of MultiCelebA in three biases setting

**MultiCelebA.** First, we mainly consider MultiCelebA in two biases setting as the dataset to evaluate debiased training algorithms. As introduced in Section 4, this dataset requires training a model to predict whether if a given face image has `high-cheekbones` or not. Each image is additionally annotated with `gender` and `age` attributes which are spuriously correlated with the target `high-cheekbones`. For MultiCelebA in three biases setting, each image is annotated with `gender`, `age`, and `mouth slightly open` attributes which are spuriously correlated with the target `high-cheekbones`.

**UrbanCars.** UrbanCars (Li et al., 2023) is a dataset created by synthesizing `background`, `co-occurring object`, and `car` to generate multi-biased images. Its task involves classifying whether an image contains `urbancars` or not.

**Multi-Color MNIST.** We consider Multi-Color MNIST dataset proposed by Li et al. (2022). Its task is to predict the digit number from an image. The digit numbers are spuriously correlated with left and right background colors, coined `left-color` and `right-color`, respectively. As proposed by Li et al., we set the proportion of bias-guiding attributes to be 99% and 95% for `left-color` and `right-color`, respectively.

**Waterbirds.** Waterbirds (Sagawa et al., 2019) is a single-bias dataset consisting of bird images. Given an image, the target is `bird-type`, *i.e.*, whether if the bird is "landbird" or a "waterbird." The biased attribute is `background-type`, *i.e.*, whether if the image contains "land" or "water." The proportion of biased attribute is set to 95%.

**CelebA.** CelebA (Liu et al., 2015) is a face recognition dataset where each sample is labeled with 40 attributes. Following the previous settings (Sagawa et al., 2019; Yao et al., 2022), we use `HairColor` as the target and `gender` as the bias attribute.

**BFFHQ.** BFFHQ (Lee et al., 2021) is a real-world face image dataset curated from FFHQ. Its task is to predict the age from an image. the age is spuriously correlated with gender attributes. The proportion of bias-guiding attributes is 99.5%.

### A.4.2 BASELINES

We extensively compare our algorithm against the existing debiased training algorithms. In particular, one can categorize a baseline by whether it explicitly uses the supervision on biased attributes, *i.e.*, bias labels, or not. To this end, compare our method with nine training algorithms, consisting of five that do not use the bias label and six that do. Algorithms that do not require using the bias label are as follows: (1) training with vanilla ERM, (2) LfF (Nam et al., 2020) employs a reweighting scheme where samples that are more likely to be misclassified by a biased model are assigned higher weights, (3) JTT (Liu et al., 2021) retrains a model using different

weights for each group, where the groups are categorized as either bias-guiding or bias-conflicting based on an ERM model, (4) EIIL (Creager et al., 2021) conducts domain-invariant learning, (5) PGI (Ahmed et al., 2021) matches the class-conditional distribution of groups by introducing predictive group invariance, and (6) DebiAN (Li et al., 2022) utilizes a pair of alternate networks to discover and mitigate unknown biases sequentially. We consider debiased training methods using bias attribute labels as follows: (1) Upsampling assigns higher sampling probability to minority groups, (2) Upweighting assigns scales the sample-wise loss to be higher for minority groups; group weight $=$ (# of training samples)$/$(# of group samples), (3) GroupDRO (Sagawa et al., 2019) computes group-scaling weights using group-wise training loss to upweight the worst-case group samples. (4) SUBG (Sagawa et al., 2020) proposes a group-balanced sampling scheme by undersampling the majority groups. (5) LISA (Yao et al., 2022) performs group mixing (mixup) augmentation to learn from both intra- and inter-group information. (6) DFR (Kirichenko et al., 2022) retrains the last layer of an ERM model using a balanced set obtained through undersampling.

Table A3: The search spaces of hyperparameters.

| Hyperparameter | Search space |
|---|---|
| Learning rate $\eta_1, \eta_2$ | {5e−4, 2e−4, 1e−4, 5e−3, 2e−3, 1e−3, 5e−2, 2e−2, 1e−2} |
| Weight decay | {0, 1e−4, 1e−2, 1e−1, 1} |
| Update frequency $U$ | {1, 5, 10, 50} |

Table A4: Hyperparameters of our method. MultiCelebA (2) represents MultiCelebA in two biases setting, and MultiCelebA (3) represents MultiCelebA in three biases setting.

| | MultiCelebA (2) | MultiCelebA (3) | Multi-Color MNIST | UrbanCars | Waterbirds | CelebA | BFFHQ |
|---|---|---|---|---|---|---|---|
| Batch size | 512 | 512 | 512 | 128 | 128 | 128 | 64 |
| Learning rate $\eta_1$ | 2e−4 | 2e−4 | 2e−2 | 1e−2 | 1e−3 | 2e−3 | 2e−3 |
| Learning rate $\eta_2$ | 2e−2 | 1e−2 | 2e−3 | 1e−3 | 1e−3 | 1e−4 | 5e−4 |
| Update frequency $U$ | 10 | 5 | 50 | 10 | 5 | 1 | 1 |
| Optimizer | SGD | SGD | Adam | SGD | SGD | Adam | Adam |

### A.4.3 HYPERPARAMETERS

We tune all hyperparameters, as well as early stopping, based on highest WORST for MultiCelebA, UrbanCars, Waterbirds and CelebA on validation set, except for ERM. For Multi-Color MNIST and BFFHQ, we tune hyperparemters based on highest UNBIASED on test set, following the previous work (Li et al., 2022; Lee et al., 2021). We use a single GPU (RTX 3090) for training. Following the previous work (Lee et al., 2021; Hwang et al., 2022), we conduct experiments on BFFHQ using ResNet18 with random initialization as the neural network architecture. For remaining datasets, we initialized the model with parameters pretrained on ImageNet. The hyperparameter search spaces used in all experiments conducted in this paper are summarized in Table A3. The selected hyperparameters for our method are represented in Table A4. Furthermore, the search space for the upweight value $\lambda_{up}$ in JTT is 5, 10, 20, 30, 40, 50, 100. JTT (Liu et al., 2021) and DFR (Kirichenko et al., 2022) utilize the ERM model as a pseudo labeler and frozen backbone network, respectively. We used the ERM model as reported in the literature for our implementation of these methods.

Given that the proportion of samples from minority groups can impact the performance of debiased training, we trained DFR exclusively on the training set to ensure a fair comparison, which is denoted as $\text{DFR}_{tr}^{tr}$.

### A.4.4 TRAINING EXISTING METHODS ON MULTI-BIAS SETTING

When training a model using SUBG (Sagawa et al., 2020), GroupDRO (Sagawa et al., 2019) and DFR (Kirichenko et al., 2022), we grouped the training set based on the same pair of bias attribute $b$ and target class $t$ and followed the approach outlined in the original paper.

LISA (Yao et al., 2022) adopts the two kinds of selective augmentation strategies, Intra-label LISA and Intra-domain LISA. In the multi-bias setting, Intra-label LISA (LISA-L) interpolates samples

with the same target label but different all bias labels ($t^{(m)} = t^{(m')}$, $b_d^{(m)} \neq b_d^{(m')}\ \forall d$). Intra-domain LISA (LISA-D) interpolates samples with the same bias labels but different target label ($t^{(m)} \neq t^{(m')}$, $\boldsymbol{b}^{(m)} = \boldsymbol{b}^{(m')}$).

When training a model using biased training methods that do not require bias labels, such as LfF (Nam et al., 2020), JTT (Liu et al., 2021), and DebiAN (Li et al., 2022), we followed the approach outlined in the original paper without modification, regardless of the number of bias types presented in the dataset.

### A.4.5 EVALUATION METRICS

We consider various metrics to evaluate whether if the trained model is biased towards a certain group in the dataset. We remark that no metric is universally preferred over others, *e.g.*, worst-group and average-group accuracy reflects different aspects of a debiased training algorithm. For the multi-bias datasets, we evaluate algorithms using the average accuracy for each of the four groups categorized by the guiding or conflicting nature of the biases: {GG, GC, CG, CC}. Here, G and C describes whether a group contains bias-guiding or bias-conflicting samples for each bias type, respectively. For example, GC group for MultiCelebA is an intersection of bias-guiding samples with respect to the first bias type, *i.e.*, gender, and bias-conflicting samples with respect to the second bias type, *i.e.*, age. We also report the average of these four metrics, denoted as UNBIASED. Next, for the single-bias datasets, the minimum group average accuracy is reported as WORST, and the weighted average accuracy with weights corresponding to the relative proportion of each group in the training set as INDIST (in-distribution) following Sagawa et al. (2019). We also report WORST and INDIST metrics on MultiCelebA.

In calculating the GG, GC, CG, CC accuracies on the MultiCelebA dataset, we excluded the impact of class imbalance within each group by first computing the mean accuracy for each class within the group, and then taking the average of the class accuracies to obtain the group accuracy.

### A.5 INTERPRETATION OF THE RESULTS ON MULTICELEBA

In Table 1, we analyzed whether a model is biased toward the two bias types, based on the difference between GG, GC, CG, CC, while also evaluating the UNBIASED accuracy. Let G* denote the combination of GG and GC, and similarly for C* and others. A model is biased towards gender attributes if there is a significant difference between the G* and C* combinations, whereas a significant difference between the *G and *C combinations indicates bias towards age attributes.

Table A5: WORST and INDIST metrics (%) evaluated on Waterbirds and CelebA. We mark the best and the second-best performance of WORST and INDIST in **bold** and underline, respectively.

| Method | Bias label | Waterbirds WORST | Waterbirds INDIST | CelebA WORST | CelebA INDIST |
|---|---|---|---|---|---|
| ERM | ✗ | $63.7_{\pm1.9}$ | $97.0_{\pm0.2}$ | $47.8_{\pm3.7}$ | $94.9_{\pm0.2}$ |
| LfF (Nam et al., 2020) | ✗ | 78.0 | 91.2 | 70.6 | 86.0 |
| EIIL Creager et al. (2021) | ✗ | $77.2_{\pm1.0}$ | $96.5_{\pm0.2}$ | $81.7_{\pm0.8}$ | $85.7_{\pm0.1}$ |
| JTT (Liu et al., 2021) | ✗ | $83.8_{\pm1.2}$ | $89.3_{\pm0.7}$ | $81.5_{\pm1.7}$ | $88.1_{\pm0.3}$ |
| LWBC (Kim et al., 2022) | ✗ | - | - | $85.5_{\pm1.4}$ | $88.9_{\pm1.6}$ |
| CNC (Zhang et al., 2022) | ✗ | $88.5_{\pm0.3}$ | $90.9_{\pm0.1}$ | $88.8_{\pm0.9}$ | $89.9_{\pm0.5}$ |
| Upweighting | ✓ | $88.0_{\pm1.3}$ | $95.1_{\pm0.3}$ | $83.3_{\pm2.8}$ | $92.9_{\pm0.2}$ |
| GroupDRO (Sagawa et al., 2019) | ✓ | $\underline{89.9}_{\pm0.6}$ | $92.0_{\pm0.6}$ | $88.9_{\pm1.3}$ | $93.9_{\pm0.1}$ |
| SUBG (Sagawa et al., 2020) | ✓ | $89.1_{\pm1.1}$ | - | $85.6_{\pm2.3}$ | - |
| SSA (Nam et al., 2022) | ✓ | $89.0_{\pm0.6}$ | $92.2_{\pm0.9}$ | $\textbf{89.8}_{\pm1.3}$ | $92.8_{\pm0.1}$ |
| LISA (Yao et al., 2022) | ✓ | $89.2_{\pm0.6}$ | $91.8_{\pm0.3}$ | $\underline{89.3}_{\pm1.1}$ | $92.4_{\pm0.4}$ |
| DFR$_{tr}^{tr}$ (Kirichenko et al., 2022) | ✓ | $90.2_{\pm0.8}$ | $97.0_{\pm0.3}$ | $80.7_{\pm2.4}$ | $90.6_{\pm0.7}$ |
| Ours | ✓ | $\textbf{91.8}_{\pm0.3}$ | $95.6_{\pm0.3}$ | $\textbf{89.8}_{\pm1.3}$ | $91.4_{\pm1.2}$ |

Table A6: UNBIASED metric (%) evaluated on BFFHQ. We mark the best and the second-best performance of UNBIASED in **bold** and underline, respectively.

| Method | Bias label | BFFHQ UNBIASED |
|---|---|---|
| ERM | ✗ | $56.2_{\pm 0.4}$ |
| HEX (Wang et al., 2018) | ✗ | $52.8_{\pm 0.9}$ |
| ReBias (Bahng et al., 2020) | ✗ | $56.8_{\pm 1.6}$ |
| LfF (Nam et al., 2020) | ✗ | $65.6_{\pm 1.4}$ |
| DisEnt (Lee et al., 2021) | ✗ | $61.6_{\pm 2.0}$ |
| SelecMix (Hwang et al., 2022) | ✗ | $71.6_{\pm 1.9}$ |
| SelecMix* (Hwang et al., 2022) | ✓ | $75.0_{\pm 0.5}$ |
| EnD (Tartaglione et al., 2021) | ✓ | $56.5_{\pm 0.6}$ |
| LISA (Yao et al., 2022) | ✓ | $65.2_{\pm 0.5}$ |
| GroupDRO (Sagawa et al., 2019) | ✓ | $\underline{85.1_{\pm 0.9}}$ |
| Ours | ✓ | $\mathbf{85.7}_{\pm 0.3}$ |

## A.6 QUANTITATIVE RESULTS ON SINGLE-BIAS SETTING

In Table A5&A6, we compare our method with existing methods on single-bias benchmarks, Waterbirds, CelebA, and Biased FFHQ (BFFHQ). Our method achieves the best WORST accuracy on Waterbirds and CelebA, and the best UNBIASED on BFFHQ, indicating that our method is effective not only for multi-bias settings but also for single-bias settings.

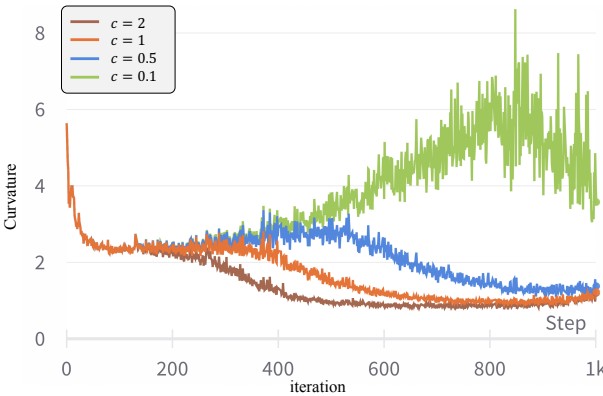

Figure A4: Local loss curvature of the loss landscape of model parameter.

## A.7 IMPACT OF THE LOSS FUNCTION ON LOCAL CURVATURE

According to Li & Gong (2021), the second term in Eq. 3, $\left\| \boldsymbol{\alpha}^\top (\nabla L(\theta)) \right\|_2^2$, serves as an approximation for the local curvature of the loss landscape associated with the model parameter $\theta$. Although this term is minimized by updating $\alpha$, the local curvature of loss landscape of model parameter is reduced. To verify this, we conducted an ablation study by adjusting the relative weight of the second term in Eq. 3 using constant $c$. The objective formula for this experiment is presented as:

$$\hat{L}(\theta) = \boldsymbol{\alpha}^\top L(\theta) + c\lambda \left\| \boldsymbol{\alpha}^\top (\nabla L(\theta))_\dagger \right\|_2^2. \tag{4}$$

Figure A.7 demonstrates how the loss curvature evolves over training iterations. We observed that as the value of $c$ decrease, there is a corresponding increase in loss curvature. Hence, minimizing the second term in Eq. 3 contributes to improving model generalization.

Table A7: Performance in UNBIASED, WORST, and INDIST (%) on MultiCelebA in three biases for evaluation two biases for training setting. We mark the best and the second-best performance in **bold** and underline, respectively.

| Method | UNBIASED | WORST | INDIST |
|---|---|---|---|
| ERM | $64.1_{\pm 0.6}$ | $12.0_{\pm 1.2}$ | $\mathbf{96.7}_{\pm 0.2}$ |
| LfF | $71.7_{\pm 0.6}$ | $47.7_{\pm 5.5}$ | $81.2_{\pm 2.9}$ |
| Upsampling | $\underline{74.1}_{\pm 1.7}$ | $48.9_{\pm 2.0}$ | $84.1_{\pm 3.0}$ |
| Upweighting | $69.7_{\pm 15.4}$ | $42.2_{\pm 24.0}$ | $79.5_{\pm 9.5}$ |
| GroupDRO | $73.7_{\pm 0.7}$ | $46.6_{\pm 0.8}$ | $83.4_{\pm 0.6}$ |
| LISA | $75.6_{\pm 1.0}$ | $\underline{52.2}_{\pm 2.0}$ | $\underline{87.3}_{\pm 0.8}$ |
| Ours | $\mathbf{78.1}_{\pm 0.6}$ | $\mathbf{58.0}_{\pm 5.4}$ | $83.8_{\pm 0.7}$ |

