# OpenReview forum: "Removing Multiple Shortcuts through the Lens of Multi-task Learning"
_ICLR.cc/2024/Conference — Submitted to ICLR 2024_

### Official Review · Reviewer_JtYi · 2023-10-31

**Soundness:** 3 good
**Presentation:** 3 good
**Contribution:** 3 good
**Rating:** 6
**Confidence:** 4

**Summary:**

This paper focuses on the problem of training an accurate unbiased model using a dataset with multiple biases. Conventional training methods (e.g. ERM) lead to undesirable shortcuts in the model due to the spurious correlations in the dataset. To counter this, debiased training algorithms have been proposed but most of them focus on a single bias at a time. Hence, this work focuses on the problem of multiple biases in a dataset, which is more practical. They first divide the dataset into multiple groups such that each group exerts the same bias on model training, using labeled bias attributes. They formulate the problem in terms of multi-task learning (MTL) where the model has to learn to handle each group correctly. Towards this, they derive a multi-objective optimization algorithm to dynamically update task weights so that model parameters can converge to a Pareto-stationary point. For experiments, they re-purpose the CelebA dataset (called MultiCelebA) using multiple attributes that are spuriously correlated with the target class. On MultiCelebA and other benchmarks, they achieve state-of-the-art performance for both multiple bias and single bias settings.

**Strengths:**

* The interpretation of debiased training as an MTL problem seems interesting and novel.

* The proposed method is simple, intuitive, and effective.

* The paper is fairly well-written and easy to follow.

**Weaknesses:**

* Debiased training as MTL
    * Debiased training actually seems closer to multi-domain learning (MDL) [W1] rather than MTL. Because MTL clearly means learning different tasks (i.e. task labels are different) as the paper also mentions on Page 2.
    * On the other hand, MDL involves the same target classes but data coming from different domains, and the goal is to achieve good performance on all domains simultaneously.
    * Also it seems like debiased training has a lot in common with long-tailed learning [W2]. For example, upweighting and upsampling baselines are also employed in long-tailed learning.
    * Overall, I wonder why MTL is chosen over these other two. Also, it might be interesting to see how more advanced techniques from MDL or long-tailed learning perform when adapted to debiased training.
    * Note: the above two references are just examples, there are many more papers in both sub-topics.

* Regarding dataset contribution
    * It is unclear how MultiCelebA is a new dataset compared to CelebA. There are no new images or new labels. Even the choice of attributes is based on the analysis from a prior work.
    * The contribution of a new dataset seems misleading, only a new experimental setting based on an existing dataset is proposed.

* Design choices
    * The design choices for the main algorithm are not well explained. Please see questions for more details.

* Practicality and significance of dataset
    * Simple methods like upsampling and upweighting give very good improvements over the ERM baseline and are quite close to even the proposed method (Table 1). This raises a question on whether the proposed MultiCelebA is practical or challenging enough to provide new insights into evaluating debiased training methods.
    * See questions for more details.

### References

[W1] Sebag et al., "Multi-Domain Adversarial Learning", ICLR19

[W2] Cao et al., “Learning Imbalanced Datasets with Label-Distribution-Aware Margin Loss”, NeurIPS19

**Questions:**

* Please also see the weaknesses section.

* Regarding practicality and significance of datasets and debiased training
    * Ideally, one would expect new datasets or benchmarks to improve over existing datasets by providing more challenging scenarios (or at least more data). While the overall accuracy numbers of MultiCelebA seem lower than UrbanCars (but similar to Multi-Color MNIST), upsampling and upweighting are consistently good on all three datasets.
    * So MultiCelebA seems similar in terms of difficulty compared to existing datasets (i.e. there seems to be no advantage to having real images over synthetic images in the other datasets).
    * Another question is whether complicated debiased training methods (like the proposed method) themselves provide any significant improvements to be deemed of practical significance. This is because we observe a similar trend where upsampling and upweighting perform very well compared to all the supervised debiased learning methods.

* Regarding design choices (Algorithm 1)
    1. Why is $\theta$ updated again (outside the "for loop") after updating it for $U$ number of times?
    2. Why do $\bar{\alpha}$ and $\lambda$ need to be updated once every $U$ iterations and not every iteration?
    3. Also, why not have $\bar{\alpha}$ and $\lambda$ updates be at different frequencies like every $U_1$ and every $U_2$ iterations instead of together after $U$ iterations?
    * These design choices need to be explained and justified since this is the core idea being proposed.

* Minor comments
    * Page 3 (5th line of “Fairness with MTL” paragraph): use \cite instead of \citet
    * Sec. 3.1 (first paragraph): mention dimensions of $\theta$, should be $\mathbb{R}^n$ as per usage in Definition 2.
    * Fig. 3: x-axis title has typos: “interation” → “iteration”.
    * Page 10: Citation for Fernando et al. (MoCo) is incorrect, it should be ICLR 2023 and not ICLR 2022.

---

> ### Author Response · Authors · 2023-11-21
> **Response to the Reviewer JtYi (1/2)**
>
> We sincerely appreciate your insightful feedback and constructive suggestions that helped improve our paper substantially. All the suggestions will be incorporated, and additional discussions will be added to the main text. Please find our detailed responses to the comments below.
> ____
> >**[Weakness 1] Debiased training as MTL: Debiased training actually seems closer to multi-domain learning (MDL) [W1] rather than MTL. Because MTL clearly means learning different tasks (i.e. task labels are different) as the paper also mentions on Page 2.**
> >- On the other hand, MDL involves the same target classes but data coming from different domains, and the goal is to achieve good performance on all domains simultaneously. Also it seems like debiased training has a lot in common with long-tailed learning [W2]. For example, upweighting and upsampling baselines are also employed in long-tailed learning.
> >- Overall, I wonder why MTL is chosen over these other two. Also, it might be interesting to see how more advanced techniques from MDL or long-tailed learning perform when adapted to debiased training.
> >- Note: the above two references are just examples, there are many more papers in both sub-topics.
>
> Thank you for interesting and insightful suggestions! As you mentioned, both long-tailed classification and multi domain learning are relevant in mitigating spurious correlation in the context of data imbalances. Based on your comments, we will consider integrating techniques from MDL or long-tailed learning for debiasing in the future work. However, we believe MTL can be more naturally incorporated with debiased training for the following reasons. Debiasing focuses on training an unbiased model with respect to spurious correlations. While MDL addresses data imbalance among different domains and long-tailed classification deals with data imbalance among target classes, debiasing tackles the imbalance among bias attributes within a given target label in training dataset. For instance, in bird species classification, if most waterbirds in the dataset are with water background and ERM erroneously uses the water background as a shortcut for classifying waterbirds, waterbirds and water background have  spurious correlation. Therefore, we designed a learning algorithm that causes task conflicts when a model relies on shortcuts by defining tasks according to adherence of samples to spurious correlations using our grouping policy. Adopting MOO to resolve task conflicts is an effective approach for training an unbiased model by removing shortcuts.
> ____
> >**[Weakness 2] It is unclear how MultiCelebA is a new dataset compared to CelebA. There are no new images or new labels. Even the choice of attributes is based on the analysis from a prior work. The contribution of a new dataset seems misleading, only a new experimental setting based on an existing dataset is proposed.**
>
> Thanks for the comment, and we have replaced “new dataset” with “new benchmark” in the manuscript to address the concern. Despite this change, we still believe that, as a realistic and challenging benchmark, MultiCelebA will contribute substantially and promote future research in the field of debiased training. We also would like to highlight that reprocessed datasets/benchmarks like MultiCelebA have long been recognized by the community; examples include bFFHQ, a dataset for debiasing derived from FFHQ, and those for long-tailed classification such as ImageNet-LT (from ImageNet), CIFAR-LT (from CIFAR), and Places-LT (from Places).

---

> > ### Author Response · Authors · 2023-11-21
> > **Response to the Reviewer JtYi (2/2)**
> >
> > >**[Weakness 3] Simple methods like upsampling and upweighting give very good improvements over the ERM baseline and are quite close to even the proposed method (Table 1). This raises a question on whether the proposed MultiCelebA is practical or challenging enough to provide new insights into evaluating debiased training methods.**
> > >
> > >**[Question 1-3] Another question is whether complicated debiased training methods (like the proposed method) themselves provide any significant improvements to be deemed of practical significance. This is because we observe a similar trend where upsampling and upweighting perform very well compared to all the supervised debiased learning methods.**
> >
> > First of all, we would like to correct an erratum in Table 1 of the main paper: the 'Unbiased' accuracy of 'Upweighting' should be 79.4, not 81.2. We deeply apologize for our mistake, and would like to confirm that it was indeed a simple erratum and we did not retrain the model to obtain a new result as follows:
> > - The 'Unbiased' accuracy, by its definition, equals the average of 'GG', 'GC', 'CG', and 'CC' scores, as described in Section 5.1 ‘Evaluation metrics’.
> > - The average of the four scores is 79.4 for 'Upweighting' in Table 1, i.e.,  (79.0+79.2+80.8+78.7)/4 = 79.4.
> > This erratum has been rectified in the revision. Again, we sincerely apologize for the mistake.
> >
> > Given the updated score, our method clearly outperformed prior arts including 'Upweighting' in Table 1 (and in the other tables too). In particular, existing debiasing methods designed for single bias settings are on par with or even underperform 'Upweighting' on MultiCelebA. These results suggest that MultiCelebA presents new challenges in the area of debiased training and that advanced debiased training algorithms are required to improve performance on the dataset.
> > ___
> > >**[Question 1-1] Ideally, one would expect new datasets or benchmarks to improve over existing datasets by providing more challenging scenarios (or at least more data). While the overall accuracy numbers of MultiCelebA seem lower than UrbanCars (but similar to Multi-Color MNIST), upsampling and upweighting are consistently good on all three datasets. So MultiCelebA seems similar in terms of difficulty compared to existing datasets (i.e. there seems to be no advantage to having real images over synthetic images in the other datasets).**
> >
> > **The difficulty of MultiCelebA compared to existing datasets**
> >
> > Due to the differences in model architectures used for datasets, a direct comparison of overall accuracy numbers not fully represents the relative difficulty of each dataset; Unlike MulticelebA and UnbarCars, where ResNet18 was used, a simple three-layed MLP was utilized in Multi-Color MNIST, resulting in lower overall accuracy numbers for Multi-Color MNIST for its difficulty.
> >
> > **MultiCelebA provide more challenging scenarios**
> >
> > MultiCelebA provides various experimental settings, including a two bias setting in Table 1, a three bias setting in Table 2, and a three bias setting with one missing bias label for training in Table A7. Additionally, it is a 8.4 times larger-scale dataset (66592 training images) compared to UrbanCars (8000 training images). This aspect is noteworthy, as illustrated in Figure 1, where the intersection of bias-conflicting samples (i.e., the smallest group) becomes extremely small as the number of bias types increases.
> > ___
> > >**[Question 2] Regarding design choices (Algorithm 1) 1. Why is $\theta$ updated again (outside the "for loop") after updating it for $U$ number of times? 2. Why do $\bar{\alpha}$ and $\lambda$ need to be updated once every $U$ iterations and not every iteration? 3. Also, why not have $\bar{\alpha}$ and $\lambda$ updates be at different frequencies like every $U1$ and every $U2$ iterations instead of together after $U$ iterations? These design choices need to be explained and justified since this is the core idea being proposed.**
> >
> > Our algorithm is designed to update $\alpha$ and $\lambda$ once every time $\theta$ is updated $U$ times. For computational efficiency, we occasionally updated $\bar{\alpha}$ and $\lambda$, and empirically demonstrated that this maintains consistent performance, regardless of updating frequency $U$. As you mentioned, we can update $\bar{\alpha}$ and $\lambda$ at different frequencies. Updating them at every $U$ iteration is a special case of updating with $U1$ and $U2$, so the performance could potentially be improved by using these distinct updating frequencies $U1$ and $U2$. We however have objections to introducing additional hyperparameters that increase the training complexity.
> > ____
> > >**[Typos]**
> >
> > We have fixed them. We appreciate your thoughtful comments.

---

> ### Comment · Reviewer_JtYi · 2023-11-23
> **Response to authors**
>
> Thanks for your efforts in the discussion period. Some of my concerns have been addressed.
>
> * However, I am still unconvinced by the argument that MTL is more suitable than MDL. I agree that either may be practically used but MDL still seems more intuitive. As per the author response, there is a task conflict when the model relies on spurious correlations, which is what MDL also tries to solve. For example, in a setup with two domains: day and night images, the spurious correlation is time of day and the model may rely on that to make the prediction. Then, it is the domain that is changing and not the task itself, the task of classification stays the same.
>
> Overall, I update my rating to "6: marginally above acceptance threshold".

---

### Official Review · Reviewer_4jDT · 2023-10-31

**Soundness:** 3 good
**Presentation:** 3 good
**Contribution:** 3 good
**Rating:** 6
**Confidence:** 3

**Summary:**

The paper effectively highlights the challenge of training models on biased datasets and the potential pitfalls of spurious correlations. The proposed method based on multi-task learning (MTL) is an innovative approach to addressing the problem of multiple biases in training data. This introduces a new perspective on debiased training. The introduction of a new real-image dataset, MultiCelebA, is a valuable contribution. It allows for evaluation under more realistic and challenging scenarios compared to existing synthetic-image datasets.

**Strengths:**

1. Application of multitask learning in the context of multiple shortcuts is a novel idea.
2.  MultiCelebA dataset can be instrumental for future research on evaluating shortcut learning algorithms.

**Weaknesses:**

Limitation:
1. No related works on shortcuts. Relevant literature can be found in
Discover and Cure: Concept-aware Mitigation of Spurious Correlation Wu et al. ICML 2023.

2. The paper is not easy to follow. The writing could have been better.

3. No code, so limited reproducibility.

4. The major issue with the approach is knowing so many different subgroups a priori. In a more challenging setting, it is almost impossible to know all possible different subgroups beforehand to design the training strategy. I would like see an experiment where out of 3 subgroups the authors include two in their training, leaving one unidentified and how their method performs.

5. There is a recent notion of difficulty in shortcuts. For example, some shortcuts are easy to learn, and some are difficult to learn. If the authors are using different subgroups to design multitask losses, the losses should be weighted corresponding to the shorcut difficulty. For example, a hard shortcut loss will be penalized less than an easy shortcut loss, as the mode is more prone to latching on the easy shotcut. The paper on shortcut difficulty is as follows:
Beyond Distribution Shift: Spurious Features Through the Lens of Training Dynamics. Murali et al. TMLR 2023.

This setting will be more realistic.

6. The dataset MultiCelebA is good for evaluation, but I would like to see a more realistic dataset like NIH-chesttube in the shortcut paper in #5.

7. Also, with multiple groups involved in the multitask loss, the overall performance may drop.

8. This is related to #4. All the possible groups may not be a shortcut. In this regard, can the shortcut discovery be aligned with the notion of slice discovery (ex DOMINO) to detect if really a spurious correlation going on before applying their method? This is a nice-to-have comment. I request the authors to think about this as a future work.


**Post rebuttal**

Thanks for the response. I agree about point 5 and 6 but request the authors to think about it. I would like to thank the authors for  considering the point 4 and would like to give a score of 7. Unfortunately, I cant assign that so i would keep my score.

**Questions:**

See the weakness

---

> ### Author Response · Authors · 2023-11-21
> **Response to the Reviewer 4jDT (1/2)**
>
> We sincerely appreciate your insightful feedback and valuable suggestions that helped improve our paper substantially. All the suggestions will be incorporated and additional experiments will be added to the main text and the supplementary. Please find our detailed responses to the comments below.
> ____
> >**[Weakness 1] No related works on shortcuts. Relevant literature can be found in Discover and Cure: Concept-aware Mitigation of Spurious Correlation Wu et al. ICML 2023.**
>
> We appreciate your thoughtful comments. We will refer to them as soon as possible.
> ____
> >**[Weakness 2] The paper is not easy to follow. The writing could have been better.**
>
> We have made revisions in line with comments from other reviewers. If there are any aspects that remain unclear or if you have any additional questions, please let us know.
> ____
> >**[Weakness 3] No code, so limited reproducibility.**
>
> In camera ready, we are committed to making our code publicly available. We understand the importance of transparency and community access in research, and we will ensure to adhere to these principles.
> ____
> >**[Weakness 4] The major issue with the approach is knowing so many different subgroups a priori. In a more challenging setting, it is almost impossible to know all possible different subgroups beforehand to design the training strategy. I would like see an experiment where out of 3 subgroups the authors include two in their training, leaving one unidentified and how their method performs.**
>
> Thank you for your constructive suggestion! We have added a new experimental setting in which only ‘Gender’ and ‘Age’ labels are visible during training on the three biases setting of MultiCelebA . As shown in Table R2 and Table A7, our method significantly outperforms existing methods. Interestingly, GroupDRO and upweighting, all of which utilize bias labels, are outperformed by LfF, a debiasing algorithm that does not use bias labels. These results indicate that our method could be effective in mitigating not only known biases but also unknown biases.
>
> **[Table R2] Performance in Unbiased, Worst, and InDist (%) on MultiCelebA in three biases for evaluation two biases for training setting. We mark the best and the second-best performance in bold and underline, respectively.**
> | Method | Unbiased | Worst | InDist |
> |----------|----------|----------|----------|
> |   ERM |  $64.1_{\pm \text{0.6}}$  |  12.0$_{\pm \text{1.2}}$  |  **96.7**$_{\pm \text{0.2}}$ |
> |   LfF  | $71.7_{\pm \text{0.6}}$  |  $47.7_{\pm \text{5.5}}$  |  $81.2_{\pm \text{2.9}}$ |
> |   Upsampling  |  $\underline{74.1}_{\pm \text{1.7}}$  |  48.9$_{\pm \text{2.0}}$  |  84.1$_{\pm \text{3.0}}$ |
> |   Upweighting  |  69.7$_{\pm \text{15.4}}$  |  42.2$_{\pm \text{24.0}}$  |  79.5$_{\pm \text{9.5}}$  |
> |   GroupDRO  | 73.7$_{\pm \text{0.7}}$  |  46.6$_{\pm \text{0.8}}$  |  83.4$_{\pm \text{0.6}}$ |
> |   LISA  | $75.6_{\pm \text{1.0}}$  |  $\underline{52.2}_{\pm \text{2.0}}$  |  $\underline{87.3}_{\pm \text{0.8}}$  |
> |   Ours  |  $\textbf{78.1}_{\pm \text{0.6}}$  |  $\textbf{58.0}_{\pm \text{5.4}}$  |  83.8$_{\pm \text{0.7}}$ |
> ____
> >**[Weakness 5] There is a recent notion of difficulty in shortcuts. If the authors are using different subgroups to design multitask losses, the losses should be weighted corresponding to the shortcut difficulty. For example, a hard shortcut loss will be penalized less than an easy shortcut loss, as the mode is more prone to latching on the easy shortcut.**
>
> According to the ICLR 2024 Reviewer Guidelines, the paper by Murali et al., published one month after our submission date, is considered concurrent work. We sincerely appreciate your diligence in bringing this to our attention and will revise the draft to discuss the paper and compare our method with it. To clarify, Murali et al. categorize spurious correlation as either benign or harmful, while our goal is to eliminate ‘shortcuts’ stemming from easy-to-learn spurious correlations. We empirically demonstrated that the bias types in MultiCelebA are such easy-to-learn spurious correlations, as shown in Section A.3.
> ____
> >**[Weakness 6] This setting will be more realistic: The dataset MultiCelebA is good for evaluation, but I would like to see a more realistic dataset like NIH-chesttube in the shortcut paper in [Weakness 5].**
>
> Thank you for your suggestion! However, the metrics used for evaluating models on the NIH-chesttube benchmark differ from those commonly used to evaluate debiasing methods, which makes direct comparisons of debiasing methods on this benchmark challenging. Currently, it is difficult to complete the experiment immediately due to the lack of time, but we aim to include it in the camera-ready if it is completed in time.

---

> > ### Author Response · Authors · 2023-11-21
> > **Response to the Reviewer 4jDT (2/2)**
> >
> > >**[Weakness 7] Also, with multiple groups involved in the multitask loss, the overall performance may drop.**
> >
> > If you could kindly provide more details regarding your comments, it would be helpful for us to address them appropriately.
> >
> > ___
> > >**[Weakness 8] This is related to [Weakness 4]. All the possible groups may not be a shortcut. In this regard, can the shortcut discovery be aligned with the notion of slice discovery (ex DOMINO) to detect if really a spurious correlation going on before applying their method? This is a nice-to-have comment. I request the authors to think about this as a future work.**
> >
> > Thank you for interesting suggestions! As discussed in Section 3.3, we can utilize existing techniques for pseudo labeling of bias attributes for grouping, effectively replacing GT bias labels. Similarly, slice discovery can be adopted for pseudo labeling within our grouping policy. Additionally, we might consider using FACTS [1], a shortcut discovery method inspired by DOMINO.
> >
> > [1] FACTS: First Amplify Correlations and Then Slice to Discover Bias, ICCV 2023

---

### Official Review · Reviewer_juPf · 2023-11-01

**Soundness:** 3 good
**Presentation:** 4 excellent
**Contribution:** 3 good
**Rating:** 6
**Confidence:** 2

**Summary:**

This paper points out the challenge of training an unbiased and accurate machine learning model using a biased dataset containing multiple biases, which lead to undesirable shortcuts during training. Connecting this problem to a multi-task learning problem, this work proposes a novel debiased training algorithm. In particular, the method optimizes both the weights and model parameters by training a single model for all tasks with a weighted sum of task-specific losses. Also, they built a new real-image multi-bias dataset (MultiCelebA) for this problem. that divide the training data into several groups based on the effects of biases on the model and define each task in MTL as solving the target problem for each group.

**Strengths:**

- This paper associates multiple biases issue in a biased dataset with a multi-task learning problem.
- The proposed new multi-bias dataset for debiased training is crucial for this area.
- Extensive experiments demonstrate the superiority of the proposed method.
- The paper is well-written and easy to understand.

**Weaknesses:**

- The relationship between multiple biases issue and multi-task learning is intriguing. However, the absence of comparisons with traditional multi-task learning (MTL) methods raises a question. If traditional MTL methods can also effectively optimize the problem, it would strengthen the connection between multiple biases issue and multi-task learning.

**Questions:**

- GradNorm [1] also adopts the gradient of loss weight to optimize the loss weight. Could you please discuss the relation and the difference between your algorithm and GradNorm? In addition, GradNorm is an important reference for this paper.
- Could you please conduct additional comparisons with traditional MTL methods to solidify the connection between multiple biases issue and multi-task learning?

[1] Chen et al. GradNorm: Gradient Normalization for Adaptive Loss Balancing in Deep Multitask Networks. In ICML 2018.

---

> ### Author Response · Authors · 2023-11-21
> **Response to the Reviewer juPf**
>
> We sincerely appreciate your insightful feedback and constructive suggestions that helped improve our paper substantially. All the suggestions will be incorporated, and additional discussions will be added to the main text. Please find our detailed responses to the comments below.
> ___
> >**[Weakness 1]  The relationship between multiple bias issue and multi-task learning is intriguing. However, the absence of comparisons with traditional multi-task learning (MTL) methods raises a question. If traditional MTL methods can also effectively optimize the problem, it would strengthen the connection between multiple biases issue and multi-task learning.**
>
> Thank you for your valuable comments! As you mentioned, the application of an existing MOO/MTL method along with our grouping policy showed improved performance compared to not adjusting group weights, as shown in Table 5, which strengthened the connection between multiple biases and MTL/MOO. However, the methods fall short of the desired level of unbiased performance. This highlights the superiority of our method in scenarios involving multiple spurious correlations.
> Specifically, in response to your comment, we have included the results of GradNorm in Table 5, as well as in the subsequent Table R1. GradNorm adjusts group weights based on a specific criteria, namely the inverse training rate. However, it has been known that reweighting task losses based on specific criteria can lead to instability [1], as demonstrated by the high standard deviation in Table W1. GradNorm improved the performance on bias-guiding samples, but it showed lower performance on bias-conflicting samples, even compared with that of (i) No adjusting grouping weights. Therefore, under our grouping policy, finding Pareto optimal with MOO is effective in training an unbiased model.
>
> **[Table R1] Comparison among different strategies for adjusting $\alpha$**
> |  | GG | GC | CG | CC | Unbiased | Worst | InDist |
>  |----------|----------|----------|----------|----------|----------|----------|----------|
> | (i) No adjusting group weights | 79.6$_{\pm \text{2.9}}$ | 80.0$_{\pm \text{2.2}}$ | 79.0$_{\pm \text{1.9}}$ | 78.4$_{\pm \text{1.3}}$ | 79.2$_{\pm \text{1.4}}$ | 70.8$_{\pm \text{2.7}}$ | 78.5$_{\pm \text{5.7}}$ |
> | (ii) MGDA | 81.6$_{\pm \text{3.5}}$ | 85.1$_{\pm \text{2.1}}$ | 80.1$_{\pm \text{1.3}}$ | 82.3$_{\pm \text{3.2}}$ | 82.3$_{\pm \text{0.4}}$ | 73.9$_{\pm \text{0.2}}$ | 82.7$_{\pm \text{3.7}}$ |
> | (iii) GradNorm | **85.9**$_{\pm \text{5.8}}$ | **86.9**$_{\pm \text{2.4}}$ | 78.1$_{\pm \text{3.5}}$ | 76.6$_{\pm \text{6.5}}$ | 81.9$_{\pm \text{0.6}}$ | 70.9$_{\pm \text{4.9}}$ | **86.5**$_{\pm \text{5.4}}$ |
> | (iv) MoCo | 81.7$_{\pm \text{1.3}}$ | 81.8$_{\pm \text{2.6}}$ | 77.2$_{\pm \text{0.9}}$ | 74.9$_{\pm \text{1.2}}$ | 78.9$_{\pm \text{1.5}}$ | 72.1$_{\pm \text{2.7}}$ | 83.8$_{\pm \text{1.6}}$ |
> | (v) Ours | 82.4$_{\pm \text{0.9}}$ | 85.1$_{\pm \text{0.4}}$ | **81.7**$_{\pm \text{0.4}}$ | **82.6**$_{\pm \text{1.0}}$ | **82.9**$_{\pm \text{0.2}}$ | **77.9**$_{\pm \text{0.2}}$ | 84.3$_{\pm \text{0.9}}$ |
>
> [1] Fernando, Heshan, Han Shen, Miao Liu, Subhajit Chaudhury, Keerthiram Murugesan, and Tianyi Chen. “Mitigating Gradient Bias in Multi-Objective Learning: A Provably Convergent Stochastic Approach.” ICLR 2023.
> ___
> >**[Question 1] GradNorm [2] also adopts the gradient of loss weight to optimize the loss weight. Could you please discuss the relation and the difference between your algorithm and GradNorm? In addition, GradNorm is an important reference for this paper.**
>
> Thank you for informing us! Our algorithm updates group weights by minimizing the weighted sum of group losses and the norm of the weighted sum of gradients to perform well on all groups, while GradNorm updates group weights towards inverse training rate so that different tasks train at similar rates. We will add GradNorm in our revised manuscript.
>
> [2] Chen, Zhao, Vijay Badrinarayanan, Chen-Yu Lee and Andrew Rabinovich. “GradNorm: Gradient Normalization for Adaptive Loss Balancing in Deep Multitask Networks”. ICML 2018.
>
> ___
> >**[Question 2] Could you please conduct additional comparisons with traditional MTL methods to solidify the connection between multiple biases issue and multi-task learning?**
>
> Please see our answer to [Weakness 1].

---

> > ### Comment · Reviewer_juPf · 2023-11-22
> > **Reviewer Response**
> >
> > Thank you for your comprehensive response, which has effectively addressed my concerns. This work deserves a score of 7, but since our rating system does not include this option, I am unable to assign it. As I am not an expert in this area, I will maintain my original score.

---

### Official Review · Reviewer_tCm5 · 2023-11-01

**Soundness:** 1 poor
**Presentation:** 3 good
**Contribution:** 2 fair
**Rating:** 5
**Confidence:** 3

**Summary:**

The authors address the challenge of mitigating multiple spurious correlations. They introduce a novel dataset splitting method and construct a multi-task learning problem based on the split dataset. Their proposed algorithm identifies a Pareto-stationary parameter within this multi-task learning setup, which then becomes the model's resultant parameter. Additionally, they created the MultiCelebA dataset to benchmark the issue of multiple spurious correlations. Experimental results show that their method surpasses existing approaches in mitigating spurious correlations across three multi-bias and three single-bias datasets.

**Strengths:**

1. The paper is articulate and well-structured, making it accessible to readers.

2. The authors provide comprehensive experimental comparisons between their approach and existing methods. These results convincingly establish that the proposed method is superior in specific aspects, underscoring its advantages.

**Weaknesses:**

1. The rationale behind the algorithm design remains ambiguous. Although the authors mention that a Pareto-stationary point with a flat loss landscape helps resolve between-group conflicts, the underlying logic is not evident. Rigorous definitions of spurious correlations and the conditions under which they are fully eliminated would benefit readers.

2. The MultiCelebA dataset appears inadequate in distinguishing between spurious and non-spurious correlations. The authors have not elaborated on the dataset's construction, even in supplementary materials. Furthermore, while they label certain correlations between target attributes and specific attributes as spurious, the these correlations seem to compose of the spurious and non-spurious ones.  An isolated evaluation of spurious correlations is essential for gauging the efficacy of methods designed to counteract them. Thus, relying on experiments with the MultiCelebA dataset might be questionable.

3. Tables 2-4 present diverse evaluation metrics, raising concerns about potential cherry-picking to favor the proposed method. Without including metrics both UNBIASED and WORST consistently, the experiments might come off as biased and not entirely objective.

4. The credibility of evaluating spurious correlations using the MultiCelebA dataset is questionable, as mentioned above. While results derived from Multi-Color MNIST might be more reliable, the proposed method's minimum group-wise accuracy is lower than that of GroupDRO. This undermines the claim that the proposed method is superior to GroupDRO.

**Questions:**

1. What underpins the expectation that the proposed method will effectively address multiple spurious correlations?

---

> ### Author Response · Authors · 2023-11-21
> **Response to the Reviewer tCm5 (1/2)**
>
> Thank you for dedicating your time and expertise to review our manuscript. Your insights are invaluable to us. We will do our best to address all of them in the revision. Please find our responses to the comments below.
> ___
> >**[Weakness 1-1] Rigorous definitions of spurious correlations and the conditions under which they are fully eliminated would benefit readers.**
>
> Thank you for your comprehensive suggestion! We will revise our manuscript to include the following content to our paper.
>
> A spurious correlation is a statistical relationship between two variables that appears to be causally related but is actually due to a coincidence. Spurious correlations appear in a majority of training data, but not the majority in testing data. For example, in bird species classification, if most waterbird images in a dataset have water backgrounds, there exists a spurious correlation between waterbird and water backgrounds. A model trained with ERM may erroneously use the water background as a shortcut for classifying waterbirds, leading to misclassification of test waterbird images with no water backgrounds. The model that learns such shortcuts is referred to as a biased model. A biased model well classifies bias-guiding samples (i.e., samples that agree with the spurious correlations) but fails to correctly classify bias-conflicting samples (i.e., those disagreeing with the spurious correlations). On the contrary, when a model learns the target task while avoiding shortcuts, it learns a feature space free from spurious correlations while representing the target task. As a result, the model exhibits unbiased accuracy between bias-guiding samples and bias-conflicting samples.
>
> >**[Weakness 1-2] The rationale behind the algorithm design remains ambiguous. Although the authors mention that a Pareto-stationary point with a flat loss landscape helps resolve between-group conflicts, the underlying logic is not evident.**
>
> In the presence of a biased training dataset, when a model relies on spurious correlations as shortcuts for target classification, it becomes a biased model. This biased model performs well when classifying bias-guiding samples but struggles with bias-conflicting samples. Our policy groups data according to their adherence to spurious correlations and defines each task for each of such groups so that, if a model is spuriously correlated with a bias type, tasks that adhere to the bias and those that do not adhere conflict with each other. These conflicts are analogous to task conflicts in MTL, and MOO has been adopted to resolve task conflicts of MTL. Following this, we proposed a new training objective based on MGDA, which is one of MOO methods, to relieve task conflicts by finding Pareto stationary which is a point where no further improvement can be made in any objective. In our scenarios, since all tasks aim to solve the same target classification, there in principle exists an optimal solution i.e., a feature space free from spurious correlations, and fits perfectly across all groups. Therefore optimization towards Pareto optimality with our grouping policy drives the model towards being unbiased.
> ___
> >**[Weakness 2] The MultiCelebA dataset appears inadequate in distinguishing between spurious and non-spurious correlations. The authors have not elaborated on the dataset's construction, even in supplementary materials. Furthermore, while they label certain correlations between target attributes and specific attributes as spurious, these correlations seem to compose of the spurious and non-spurious ones. An isolated evaluation of spurious correlations is essential for gauging the efficacy of methods designed to counteract them. Thus, relying on experiments with the MultiCelebA dataset might be questionable.**
>
> We would like to kindly note that Section A.3 and Figure A3 empirically demonstrated that the bias types in MultiCelebA are spuriously correlated to the target class, High Cheekbones. Additionally, Figure 2 and Table A1&A2 present statistics of MultiCelebA. If you have already reviewed these sections and find any aspect unclear or have specific concerns, we would greatly appreciate further clarification on which parts you found problematic.

---

> > ### Author Response · Authors · 2023-11-21
> > **Response to the Reviewer tCm5 (2/2)**
> >
> > >**[Weakness 3] Tables 2-4 present diverse evaluation metrics, raising concerns about potential cherry-picking to favor the proposed method. Without including metrics both UNBIASED and WORST consistently, the experiments might come off as biased and not entirely objective.**
> >
> > In Table 2, we presented the results of previous methods with respect to both UNBIASED and WORST metrics. However, for Table 3 and Table 4, we followed the evaluation protocols used in prior studies [1, 2] for UrbanCars and Multi-Color MNIST to ensure a fair comparison, which led to the absence of certain metrics in Table 3 and Table 4.
> >
> > [1] Zhiheng Li, Ivan Evtimov, Albert Gordo, Caner Hazirbas, Tal Hassner, Cristian Canton Ferrer, Chenliang Xu, and Mark Ibrahim. A whac-a-mole dilemma: Shortcuts come in multiples where mitigating one amplifies others. In Proc. IEEE Conference on Computer Vision and Pattern Recognition (CVPR), pp. 20071–20082, 2023.
> >
> > [2] Zhiheng Li, Anthony Hoogs, and Chenliang Xu. Discover and mitigate unknown biases with debias- ing alternate networks. In Proc. European Conference on Computer Vision (ECCV), pp. 270–288. Springer, 2022.
> > ___
> > >**[Weakness 4-1] The credibility of evaluating spurious correlations using the MultiCelebA dataset is questionable, as mentioned above.**
> >
> > Please see our answer to [Weakness 2].
> >
> > >**[Weakness 4-2] While results derived from Multi-Color MNIST might be more reliable, the proposed method's minimum group-wise accuracy is lower than that of GroupDRO. This undermines the claim that the proposed method is superior to GroupDRO.**
> >
> > Our method outperforms GroupDRO on UrbanCars, MultiCelebA, Waterbirds, and bFFHQ. This underscores the effectiveness of our framework, showcasing superior performance in both multiple bias scenarios and conventional single bias settings.
> >
> > We recognize that neither GroupDRO nor our method is better than the other overall on Multi-Color MNIST. While GroupDRO achieved the highest CC accuracy on Multi-Color MNIST, it exhibited lower performance in GG, GC, CG, Unbiased accuracy compared with Ours.
> >
> > It is important to note that the experiments on Multi-Color MNIST were conducted using a three-layered MLP, whereas ResNet18 was employed for UrbanCars and MultiCelebA. Since GroupDRO is designed to improve worst group accuracy and is particularly effective when the complexity of model parameters is strongly regularized, it could achieve the best CC on Multi-Color MNIST. However, it tends to underperform when serving more complex networks than a simple MLP, as demonstrated on  UrbanCars and MultiCelebA.
> > ___
> > >**[Question 1] What underpins the expectation that the proposed method will effectively address multiple spurious correlations?**
> >
> > In multiple bias scenarios, some samples may align with one spurious correlation but may conflict with another. The common strategy in single bias scenarios, which involves upweighting the worst group or clean samples that disagree with all the spurious correlations, underperforms in the multiple bias scenarios, as shown in Table 1 & 2 & 3. Since mitigating one bias often exacerbates another in the multiple bias scenarios [3], it is important to simultaneously mitigate multiple spurious correlations. Our method defines tasks according to adherence of samples to spurious correlations. When a model relies on shortcuts from spurious correlations, these are exposed as task conflicts. By optimizing to resolve these task conflicts simultaneously, we effectively eliminate multiple spurious correlations.
> >
> > [3] Li, Zhiheng, Ivan Evtimov, Albert Gordo, Caner Hazirbas, Tal Hassner, Cristian Canton Ferrer, Chenliang Xu, and Mark Ibrahim. “A Whac-A-Mole Dilemma: Shortcuts Come in Multiples Where Mitigating One Amplifies Others.” CVPR 2023.

---

### Official Review · Reviewer_B9DR · 2023-11-01

**Soundness:** 2 fair
**Presentation:** 3 good
**Contribution:** 2 fair
**Rating:** 3
**Confidence:** 4

**Summary:**

This work addresses the problem of avoiding that models use _several_ biases (or spurious correlations) from the data to perform predictions, whereas previous work has focused in avoiding a _single_ bias. To this end, the authors relate the problem of debiasing a model with multitask learning (MTL) and proposed to group the dataset into different groups/tasks based on their proposed grouping criteria. Then, they propose to train the MTL model by introducing learnable convex task weights that are regularized to reduce the norm of the loss gradient. Finally, the authors introduce a dataset based on CelebA that contains multiple biases, and use it to compare their proposed approach with a number of previous methods. Moreover, the authors also compare their method in different single-bias experiments and ablate the different components of the proposed solution.

**Strengths:**

- The problem of addressing several biases in the dataset (rather than a single one) is interesting, and a sensible middle-ground between single-bias settings and settings with no annotations.
- The proposed approach to divide the dataset into groups is also pretty interesting and novel, to the best of my knowledge.
- The paper is well-written and easy to follow.
- The empirical results are quite positive, and they also shed light on the behaviour of existing methods when multiple biases are present in the data.

**Weaknesses:**

- W1. While MultiCelebA is interesting and useful, selling it as a "new dataset" is too much of a stretch for my taste.
- W2. Saying that this is the first work connecting "unbiasing" with MOO or MTL (which is quite freely interpreted) is arguable at best. First, one could argue that even importance-weighting approaches are already interpreting the problem as MOO, but there are even works such as FairGrad [4] that connect biases (this one, in the context of fairness, which does not necessarily imply data-imbalance) which adaptively scale gradients.
- W3. Statements about MOO and Pareto Optimality in the manuscript makes me worry about whether the authors have fully understood these concepts. For example:
  - I don't understand what it means to "address spurious correlations based on a theory of MOO".
  - "Finding _the_ Pareto-optimal parameter". There are _many_ Pareto-optimal parameters.
  - The goal is that "performance should not be biased towards a certain group". Pareto-optimality does not guarantee this. Indeed, MGDA is known to be biased towards tasks with low magnitudes. The concept the authors refer to is known as "task impartiality" in MTL (see, e.g., [1, 2]).
  - Saying that MoCo is the SotA of MOO is quite a stretch to say the least.
- W4. The arguments towards "finding a flat optima" are rather hand-wavy and unconvincing.
- W5. Related with W3, it is quite unclear to me what makes the proposed method work at all:
  - MGDA minimizes in each iteration the regularizer in Eq. 3, and it is biased towards dominated tasks (which is observed in Table 5). However, the proposed approach (which is an interpolation between ERM and MGDA, similar to CAGrad [3]) works well. Is it the interpolation? Or learning $\alpha$ using along the parameters?
  - It is rather intriguing that the grouping policy does not work well on its own. In principle, I don't see why the grouping and the training approach should not be independent.
- W6. Experiments lack statistics like standard deviations, making the effectiveness of some design choices (e.g. the update frequency $U$) quite unclear.
- W7. The results in UrbanCars are quite different from those that can be found in other works. Just as an example, the worst variant of ERM recorded in [Papers with code](https://paperswithcode.com/sota/out-of-distribution-generalization-on) has a gap of -15.4, while the one reported in the paper is of -69.2 (worse than any result of the PwC table).

[1] Javaloy, A., & Valera, I. (2022). RotoGrad: Gradient Homogenization in Multitask Learning. ICLR.

[2] Liu, L., Li, Y., Kuang, Z., Xue, J., Chen, Y., Yang, W., ... & Zhang, W. (2021). Towards impartial multi-task learning. ICLR.

[3] Liu, Bo, et al. "Conflict-averse gradient descent for multi-task learning." Advances in Neural Information Processing Systems 34 (2021): 18878-18890.

[4] Maheshwari, G., & Perrot, M. (2022). Fairgrad: Fairness aware gradient descent. arXiv preprint arXiv:2206.10923.

**Questions:**

-Q1. When using MGDA in Table 5, does it mean that $\alpha$ is tuned as in Eq. 3 but only with the regularization? Or is $\alpha$ fully solved in each iteration?

---

> ### Author Response · Authors · 2023-11-16
> **Response to the Reviewer B9DR (1/4)**
>
> Thank you for dedicating your time and expertise to review our manuscript. Your insights are invaluable to us. However, it appears that some aspects of the review may have been based on certain confusions regarding our methodology and experimental results. We are eager to address these points in the rebuttal, and will accordingly revise our paper for improving its clarity.
> ___
> >**[Weakness 1] While MultiCelebA is interesting and useful, selling it as a "new dataset" is too much.**
>
> Thank you for your constructive suggestion! In response to your comment, we have replaced "new dataset" with "new benchmark" in the manuscript. Despite this change, we still believe that, as a realistic and challenging benchmark, MultiCelebA will contribute substantially and promote future research in the field of debiased training. We also would like to highlight that reprocessed datasets/benchmarks like MultiCelebA have long been recognized by the community; examples include bFFHQ, a dataset for debiasing derived from FFHQ, and those for long-tailed classification such as ImageNet-LT (from ImageNet), CIFAR-LT (from CIFAR), and Places-LT (from Places).
> ___
>
> >**[Weakness 2] Saying that this is the first work connecting "unbiasing" with MOO or MTL is arguable. One could argue that even importance-weighting approaches (ex. FairGrad) are already interpreting the problem as MOO.**
>
> Thank you for your valuable and insightful feedback! We recognize the widespread use of reweighting methods like GroupDRO in the field. It is important to note that none of such prior work, including GroupDRO, explicitly relates their frameworks with MOO/MTL. However, our work explicitly frames debiasing from a perspective of MOO for the first time, which enables us to develop a new and effective algorithm based on recent advances in MOO. Our algorithm substantially outperformed standard reweighting techniques, and we believe its outstanding performance highlights the impact of viewing debiasing through a lens of MOO explicitly.
>
> Regarding FairGrad (TMLR 2023): According to the ICLR 2024 Reviewer Guidelines, FairGrad, published one month before our submission date, is considered as concurrent work. However, we sincerely appreciate your diligence in bringing this to our attention, and have revised the draft to discuss the paper and compare our method with it. Moreover, we would like to clarify that FairGrad focuses on grouping based on sensitive labels for fairness, a notable distinction from our work that aims to remove spurious correlations by MOO. Despite the existence of FairGrad, we thus assert that our paper is still the first to introduce MOO for mitigating spurious correlations.
> ___
>
> >**[Weakness 3-1]  I don't understand what it means to "Finding the Pareto-optimal parameter" and "address spurious correlations based on a theory of MOO". There are many Pareto-optimal parameters. And, Pareto-optimality does not guarantee that not being biased towards a certain group.**
>
> Thank you for your perceptive observation. You correctly note that, in typical MTL settings, Pareto optimality does not necessarily guarantee unbiased performance across groups: there is a trade-off between tasks in terms of performance due to the conflicts between them (i.e., different tasks have different goals), and consequently a Pareto optimal model may be biased towards a certain task. However, in our setting, since tasks share exactly the same objective, a single model that works for all the tasks exists in principle (i.e., a model free from spurious correlations); “the” Pareto optimal parameter indicates such a single, ideal model in our paper. We will clarify this in the draft soon!
>
> >**[Weakness 3-2] MGDA is known to be biased towards tasks with low magnitudes.**
>
> MGDA tends to be biased towards tasks with low magnitudes as commented, and such a behavior is in fact beneficial for our framework. We first would like to note that, as demonstrated in Section A.2, our grouping strategy and group-wise loss result in the minority group (CC) with significantly low loss magnitude. This is mainly because (1) all the groups contribute to training equally, regardless of their population, through the group-wise loss, and (2) it is easy to reduce the loss for the minority group due to the low complexity of its data distribution. Hence, MGDA allows our algorithm to focus more on the minority group (with low loss magnitudes) during training and consequently resolve the data imbalance issue between the groups.
>
> >**[Weakness 3-3] Saying that MoCo is the SOTA of MOO is quite a stretch.**
>
> We agree, and have revised the draft to refer to MoCo as "the latest technique for MOO" instead.

---

> ### Author Response · Authors · 2023-11-16
> **Response to the Reviewer B9DR (2/4)**
>
> >**[Weakness 4] The arguments towards "finding a flat optima" are rather hand-wavy and unconvincing.**
>
> This statement does not imply that a flat minima leads to an unbiased model. Our intention in the paper was that the second term in Eq. (3), which is originally introduced to achieve Pareto stationarity, can be also interpreted as an approximate curvature of the loss, according to previous work [1].
> To be specific, the trace norm of hessian tr(H) is known to represent loss curvature, and the trace norm of the empirical Fisher information $tr(F)=\mathbb{E}\left[\left\lVert \boldsymbol{\alpha}^{\top}(\nabla L(\theta))\right\rVert_2^2\right]$, which is equivalent to the second term in Eq. (3), approximates $tr(H)$ [2, 3]. Therefore, minimizing the second term in Eq. (3) encourages to find a flat optima. Also, such a flat optima has been known to improve models’ generalization capability [1, 4, 5, 6].
> To further understand the impact of the term on the loss curvature, **we have included a new ablation study in Section A.7.** The experimental results suggest that a higher weight on this term leads to a smaller loss curvature in the model parameter space.
>
> [1] Xian Li and Hongyu Gong. Robust optimization for multilingual translation with imbalanced data. In Proc. Neural Information Processing Systems (NeurIPS), volume 34, 2021.
>
> [2] Stanislaw Jastrzebski, Devansh Arpit, Oliver Astrand, Giancarlo Kerg, Huan Wang, Caiming Xiong, Richard Socher, Kyunghyun Cho, and Krzysztof Geras. Catastrophic ﬁsher explosion: Early phase ﬁsher matrix impacts generalization. arXiv preprint arXiv:2012.14193, 2020.
>
> [3] Yu Sun, Shuohuan Wang, Yukun Li, Shikun Feng, Xuyi Chen, Han Zhang, Xin Tian, Danxiang Zhu, Hao Tian, and Hua Wu. Ernie: Enhanced representation through knowledge integration. arXiv preprint arXiv:1904.09223, 2019.
>
> [4] Nitish Shirish Keskar, Dheevatsa Mudigere, Jorge Nocedal, Mikhail Smelyanskiy, and Ping Tak Pe- ter Tang. On large-batch training for deep learning: Generalization gap and sharp minima. In Proc. International Conference on Learning Representations (ICLR), 2017.
>
> [5] Gintare Karolina Dziugaite and Daniel M Roy. Computing nonvacuous generalization bounds for deep (stochastic) neural networks with many more parameters than training data. arXiv preprint arXiv:1703.11008, 2017.
>
> [6] Yiding Jiang, Behnam Neyshabur, Hossein Mobahi, Dilip Krishnan, and Samy Bengio. Fantastic generalization measures and where to find them. In Proc. International Conference on Learning Representations (ICLR), 2020.

---

> ### Author Response · Authors · 2023-11-16
> **Response to the Reviewer B9DR (3/4)**
>
> >**[Weakness 5-1] MGDA minimizes in each iteration the regularizer in Eq. 3, and it is biased towards dominated tasks (which is observed in Table 5). However, the proposed approach works well.**
>
> First of all, we would like to correct potential misunderstandings: The records of MGDA in Table 5 suggest that it works well for the minority task (CC) **but not for the dominant task (GG)**. Please note that G and C indicate whether a group includes bias-guiding or bias-conflicting samples for each bias type, respectively, as defined in Section 5.1. This is also discussed in our answer to [Weakness 3-2].
>
> >**[Weakness 5-2] Is the proposed method an interpolation between ERM and MGDA, similar to CaGrad? Or learning $\alpha$ using the parameter?**
>
> Our objective is not an interpolation between ERM and MGDA since both the first and the second terms of Eq. (3) involve $\alpha$. Furthermore, $\alpha$ is not a scalar weight but a learnable vector, enabling dynamic adjustment of group-wise importance during training. This is a significant departure from CAGrad, which employs a constant $c$ for interpolation of ERM and MGDA. Please check out Algorithm 1 in the paper, which clearly illustrates how our method utilizes $\alpha$ in a fundamentally different way from interpolation.
>
> >**[Weakness 5-3] It is rather intriguing that the grouping policy does not work well on its own. Why should the grouping and the training approach not be independent?**
>
> We would like to emphasize that our grouping policy is not designed to operate independently. Its purpose is dedicated to facilitating the effectiveness of our training algorithm in mitigating spurious correlations. Our policy groups data according to their adherence to spurious correlations and defines each task for each of such groups so that, if a model is spuriously correlated with a bias type, tasks that adhere to the bias and those that do not adhere conflict with each other. Then mitigating such conflicts by MOO resolves the spurious correlations and leads to an unbiased model consequently. Therefore, the grouping policy and the training strategy are inherently interconnected in our algorithm, each playing a vital role in addressing spurious correlations, as empirically demonstrated in Table 5 and 6 of the paper.
>
> >**[Weakness 5-4] It is unclear what makes the proposed method work.**
>
> The key ideas in our method involve framing the mitigation of spurious correlation as MTL and adapting MOO for debiasing. Specifically, due to the following reason, achieving Pareto optimality under our grouping policy aligns that the model learns a feature space free from spurious correlations.
> 1. There is an optimal solution that fits perfectly across all groups since they all share the same objective. (Please see our answer to [Weakness 3-1].)
> 2. If a model falls into shortcuts during training, the model’s performance will be reduced on certain groups, which deviates from Pareto optimality in the context of our grouping policy.
> 3. Hence, optimization towards Pareto optimality with our grouping policy drives the model towards being unbiased.

---

> > ### Author Response · Authors · 2023-11-16
> > **Response to the Reviewer B9DR (4/4)**
> >
> > >**[Weakness 6] Experiments lack statistics like standard deviations, making the effectiveness of some design choices (e.g. the update frequency U) quite unclear.**
> >
> > Thank you for your valuable comment. We have revised the draft so that now all the tables except Table 3 report both mean and standard deviation values of three trials; the updated tables still demonstrate the superiority of our method.
> > Please note that we could not report the standard deviations in Table 3 since the table was sourced directly from Li et al. [7], in which the standard deviations are not available.
> >
> > [7] Li, Zhiheng, Ivan Evtimov, Albert Gordo, Caner Hazirbas, Tal Hassner, Cristian Canton Ferrer, Chenliang Xu, and Mark Ibrahim. “A Whac-A-Mole Dilemma: Shortcuts Come in Multiples Where Mitigating One Amplifies Others.” CVPR 2023.
> >
> > ___
> > >**[Weakness 7] The results in UrbanCars are quite different from those that can be found in other works. Just as an example, the worst variant of ERM recorded in Papers with code has a gap of -15.4, while the one reported in the paper is of -69.2 (worse than any result of the PwC table).**
> >
> > We would like to clarify that there is no such discrepancy in the results. The -15.4 performance gap referenced from Papers with Code pertains to scenarios assuming a single bias, specifically, background bias in UrbanCars. In contrast, the results we have reported in our paper are the performance under multiple bias conditions (BG+CoObj gap).
> > Moreover, Table 3 in our manuscript was directly sourced from Table 5 of the paper that originally introduced UrbanCars. The results correspond perfectly with the BG+CoObj performance metrics listed in Papers with Code.
> > ___
> > >**[Question 1] When using MGDA in Table 5, does it mean that it is tuned as in Eq. 3 but only with the regularization? Or is it fully solved in each iteration?**
> >
> > For MGDA in Table 5, we updated $\alpha$ with only the regularization term, $\left\lVert \boldsymbol{\alpha}^{\top}(\nabla L(\theta))\right\rVert_2^2$.

---

> > > ### Comment · Reviewer_B9DR · 2023-11-20
> > >
> > > Dear authors, thanks for the detailed rebuttal, and excuse me for the late reply. A few points to follow up from your response:
> > >
> > > **[Weakness 5-1 and 5-4]** I still believe the term Pareto-optimality is not fully clear: it does not give any guarantee of the ones claimed in the manuscript and rebuttal. One solution could perfectly fit a group while completely neglecting another, and it would still be Pareto optimal. Indeed, any point in the Pareto front (which can go from balanced solutions to unbalanced ones) is Pareto optimal.
> > >
> > > **[Weakness 5-1]** Thanks for the clarification, there were no misunderstanding. I said dominated tasks (CC), not dominating tasks (GG). The answer **[Weakness 3-2]** rather clarified this point and **[Weakness 5-3]**.
> > >
> > > **[Weakness 5-2]** I still do believe your method to tune $\alpha$ is an interpolation between solving ERM (first term) and MGDA (second term), controlled by $\lambda$ (instead of $c$ in CAGrad).
> > >
> > > **[Question 1]** Good, all clear. However, this detail should also be clarified in the manuscript, as MGDA solves the optimization problem in each iteration.

---

> ### Author Response · Authors · 2023-11-21
>
> We sincerely thank you for your response! Based on your comments, we will add the details in the manuscript and address remaining concerns below. We hope this response effectively resolves the remaining concerns.
> ___
> >**[Weakness 5-2] I still do believe your method to tune $\alpha$ is an interpolation between solving ERM (first term) and MGDA (second term), controlled by $\lambda$ (instead of c in CAGrad).**
>
> Firstly, we would like to clarify that the first term in Eq. (3) is not the ERM but rather the weighted sum of group-wise losses. In contrast to conventional multi-task scenarios that compute all task losses from a single input, we group the dataset and calculate a task (group) loss within the group to which the input belongs. Therefore, the sum of group-wise losses is not equivalent to ERM in our grouping. ERM is a special case of the first term when $\alpha_i$ is proportional to the number of samples in the $i$-th group since the number of samples varies in each group. The first term becomes ERM on MultiCelebA when $\alpha_i$ is $\frac{\text{the number of samples in } i\text{-th group}}{\text{the number of samples in the training set}}$. However, since $\alpha$ is initialized as $[\frac{1}{(\text{the number of groups)}}, …, \frac{1}{(\text{the number of groups)}}]$ and dynamically updated to minimize Eq. (3), the first term is hardly considered as ERM.
>
> Furthermore, we update $\alpha$ by minimizing interpolation with **a learnable parameter** $\lambda$ between **the weighted sum of group-wise losses** and MGDA. The weighted sum of group-wise losses serves as a prior for increasing the group weight of the CC group, as detailed in Section A.2. As training progresses and the $\lambda$ increases, this prior intensifies in the early stages of training, and the impact of MGDA gradually strengthens.
>
> We would like to note that CAGrad updates model parameters by minimizing the interpolation between ERM and MGDA, whereas our algorithm minimizes the weighted sum of group-wise losses without MGDA for updating model parameters.
>
> ___
> >**[Weakness 3-1 and 5-4] I still believe the term Pareto-optimality is not fully clear: it does not give any guarantee of the ones claimed in the manuscript and rebuttal. One solution could perfectly fit a group while completely neglecting another, and it would still be Pareto optimal. Indeed, any point in the Pareto front (which can go from balanced solutions to unbalanced ones) is Pareto optimal.**
>
> We would like to illustrate this concept with an example. Assume that the task is to classify images in MultiCelebA as either having high cheekbones or not. We categorize the dataset into groups using our grouping policy. All groups have the common training objective of classifying images based on high cheekbones. When considering training a group, the solution space for groups with bias-guiding images can be broader than that of the bias-conflicting groups because they can exploit the spurious correlations to solve the task. The simple example is as follows:
>
> $\text{Solution}\_{GG}=${$ \theta_{\text{high cheekbones}}, \theta_{\text{gender}}, \theta_{\text{age}}$}
>
> $\text{Solution}\_{GC}=${$\theta_{\text{high cheekbones}}, \theta_{\text{gender}}$}
>
> $\text{Solution}\_{CG}=${$\theta_{\text{high cheekbones}}, \theta_{\text{age}}$}
>
> $\text{Solution}\_{CC}=${$\theta_{\text{high cheekbones}}$}
>
> In this example, $\theta\_{\text{high cheekbones}}$ is the optimal solution that performs well on all groups. $ \theta_{\text{gender}}$ or $\theta_{\text{age}}$ (i.e., the solutions that perfectly fit a group while completely neglecting the other) are not a Pareto optimal because while they decrease the group-wise losses only for bias-guiding groups and increase the group losses for bias-conflicting groups, there exists a better solution $\theta\_{\text{high cheekbones}}$ that decreases the groups losses for all groups.
> ___
> >**[Weakness 5-1], [Question 1]**
>
> Thank you! We will add the details in the manuscript.

---

> > ### Comment · Reviewer_B9DR · 2023-11-22
> >
> > Dear authors, thanks for your answer, but I am afraid they did not address my concerns. Namely:
> >
> > **[Weakness 3-1 and 5-4]** Of course, those solutions are not Pareto-optimal because there exists another one that dominates them. But that is all under the hypothesis in your example that there exists a solution that dominates all of them, which is _far_ from the spirit of multi-objective optimization. In MOO, there are plenty of Pareto-optimal solutions (those points not Pareto-dominated by any other) which form the Pareto front. If your assumption is that the Pareto front of your model is composed of a _single_ point (i.e., your model can fit all the losses perfectly), this should be stated clearly from the very first page. Now:
> >   - If this is the case, then _any_ linear combination of losses (static or dynamic) will yield that one optimum if the optimization works properly (since the hyperplane composed by the task weights always intersects with the same point in the Pareto front, see e.g. Boyd's book).
> >   - If that is not the case, then there exists many Pareto optima that yield different trade-off solutions between the groups, and the balanced solution you seek is just one of them.
> >
> > Thanks again for the response.

---

> > > ### Author Response · Authors · 2023-11-23
> > >
> > > We deeply appreciate your response and thoughtful comments! The reviewer’s discussion has greatly contributed to strengthening our paper. We will revise our manuscript to include the following details. We hope this response resolves the remaining concerns.
> > >
> > > >**But that is all under the hypothesis in your example that there exists a solution that dominates all of them**
> > >
> > > Yes exactly! A solution that dominates all the other solutions (which include solutions adopting spurious correlations) ideally always exists in our setting since we aim to solve the single classification task and minimize the same loss function across all the tasks (i.e., groups). Of course, we here assume that a model trained with our objective does not memorize the minority group to achieve Pareto optimality, in particular, due to the second term of our objective that leads to a flat minimum of loss landscape. We indeed admit that this detail should be discussed in the manuscript, and will add it to the revision as soon as possible, particularly in the first section as you commented.
> > >
> > > >**If this is the case, then any linear combination of losses (static or dynamic) will yield that one optimum if the optimization works properly (since the hyperplane composed by the task weights always intersects with the same point in the Pareto front, see e.g. Boyd's book).**
> > >
> > > We conjecture that the statement does not hold when we optimize DNNs, which are highly non-convex and updated iteratively by, e.g., stochastic gradient descent. Moreover, for the proposition to be held, the optimization process has to be independent of $\alpha$ as it states the existence of one optimum for “any” linear combination of group-wise losses, but the optimization is hardly independent of $\alpha$: to be independent, different gradients of different group-wise losses have to indicate the same direction at every iteration of optimization, which is in general infeasible since the optimal loss gradients for different groups are not the same (e.g., for the GG group, the loss gradient to fit the model towards the ‘age’ attribute is optimal, but not for the CC group). Hence, we argue that the proposition does not hold in our problem setting and that different combinations of group-wise losses will lead to different results.
> > >
> > > We acknowledge the importance of appropriately updating group weights and model parameters in each iteration when conducting MOO for DNNs, as supported by MOO for DNNs methods [1, 2, 3, 4, 5]. Achieving the Pareto Optimal in DNNs presents a complex challenge, even in scenarios where a single point exists in Pareto Optimal. **Subsequently, we comprehend the process of our method in DNNs as follows.**
> > >
> > > We iteratively update group weights $\alpha$ by minimizing Eq. (3) to reduce the risk of model parameters $\theta$ falling into suboptimal solutions during multiple iterations and guide them towards Pareto optimal over numerous iterations, along with updating model parameters with weighted sum of group-wise losses. In debiasing scenarios, since the number of samples in the CC group is extremely small, the CC group is easily neglected by ERM. Hence, simply upweighting the CC group is not enough to train an unbiased model achieving high accuracy across overall groups, but it is moderately effective to train a model to avoid shortcuts. With this prior, our algorithm is designed to increase $\alpha\_{CC}$ by optimizing the first term in Eq. (3) (the detailed mechanism of the algorithm in this regard have been discussed in Section A.2 and Figure A2), and to strengthen the impact of the second term (MGDA) of Eq. (3) gradually as training progresses ($\lambda$ is updated by gradient ascent for this purpose as described in Section 3.2.2). As the samples in the CC group are free from all shortcuts, emphasizing the weight of this group also decreases the group-wise loss of the other groups as well as that of the CC group. Hence, in early stages of training, training mainly with the first term in Eq. (3) leads to a model moderately trained across all groups. Then, in later stages of training, the second term of Eq. (3) is strengthened so that the model is further improved by optimization towards Pareto optimality.
> > >
> > >
> > > [1] Jean-Antoine Désidéri. Multiple-gradient Descent Algorithm (MGDA) for Multi-objective Optimization. Comptes Rendus Mathematique, 350(5-6), 2012.
> > >
> > > [2] Yu, Tianhe, et al. "Gradient surgery for multi-task learning.” NeurIPS 2020.
> > >
> > > [3] Suyun Liu and Luis Nunes Vicente. The Stochastic Multi-gradient Algorithm for Multi-objective Optimization and its Application to Supervised Machine Learning. Annals of Operations Research, pages 1–30, 2021.
> > >
> > > [4] Liu, Bo, et al. "Conflict-averse gradient descent for multi-task learning." NeurIPS 2021.
> > >
> > > [5] Fernando, Heshan Devaka, et al. "Mitigating gradient bias in multi-objective learning: A provably convergent approach." ICLR 2023.

---

> ### Author Response · Authors · 2023-11-23
>
> Finally, we sincerely thank you for your dedication and providing generous suggestions and comments until the end of the author-reviewer discussion period. They are indeed of great help that improves our work substantially!

---

> > ### Comment · Reviewer_B9DR · 2023-11-23
> >
> > Dear authors,
> > I am glad that my comments are of help, and that we have arrived to an agreement of the underlying assumptions in the manuscript.
> >
> > I am going to keep my score, and I will let the AC weigh my review when making a decision.
> >
> > The main reason I keep my score is that the argument of the paper after the rebuttal have shifted from MOO (as traditionally understood), to the training dynamics of a MOO problem which has a single Pareto-optimal point (which could sound not too multi-objective). This is a fundamentally different (and interesting!) topic, but far from the one discussed in the paper, as now we are talking about optimization issues, and not about choosing an optimum point of a vector-valued loss.
> >
> > Again, whether to accept or reject the paper comes down to how much importance one gives to these details, and I am glad of having a fruitful discussion with you.

---

### Meta-Review · Area_Chair_Lc4h · 2023-12-15

**Metareview:**

The paper, aiming to address multiple biases in datasets, receives mixed feedback from reviewers, with positive remarks on the problem's relevance, the paper's overall clarity, and the effectiveness of the proposed method. However, substantial concerns raised by reviewers, both before and after the rebuttal phase, contribute to the recommendation for rejection.
One positive aspect highlighted is the interest in addressing multiple biases in datasets, showcasing the paper's potential significance. The authors are credited for presenting their work in a well-written and easy-to-follow manner, emphasizing the simplicity, intuitiveness, and effectiveness of the proposed method. Comprehensive experimental comparisons are acknowledged for convincingly establishing the superiority of the proposed approach in specific aspects.
On the negative side, there are critical concerns about the characterization of MultiCelebA as a "new dataset," perceived as an exaggeration by one reviewer. The absence of statistical information, such as standard deviations, in the experimental results raises doubts about the robustness of the presented findings. Ambiguities in the rationale behind the algorithm design, coupled with concerns about the MultiCelebA dataset's ability to distinguish between spurious and non-spurious correlations, contribute to a lack of confidence in the paper's theoretical foundations.
Specific critiques about Tables 2-4 presenting diverse evaluation metrics, potential cherry-picking, and challenges in understanding the paper's content and design choices further contribute to the negative evaluation. The post-rebuttal comment from a confident reviewer raises fundamental concerns about the authors' understanding of Pareto-optimality and underlying assumptions, adding to the skepticism about the technical contributions.
Despite positive aspects, the substantial issues related to dataset characterization, experimental methodology, and theoretical foundations collectively contribute to the decision to reject the paper. I encourage the authors to submit an improved version to the next venue by incorporating the feedback.

**Justification For Why Not Higher Score:**

Though few reviewers moved towards borderline accept, one confident reviewer had serious reservations regarding in accepting the paper.

**Justification For Why Not Lower Score:**

N/A

---

### Decision · Program_Chairs · 2024-01-16

Reject